# Non-Hermitian non-equipartition theory for trapped particles

Xiao Li [1,2,4], Yongyin Cao [3,4] & Jack Ng [1]

The equipartition theorem is an elegant cornerstone theory of thermal and statistical physics. However, it fails to address some contemporary problems, such as those associated with optical and acoustic trapping, due to the non-Hermitian nature of the external wave-induced force. We use stochastic calculus to solve the Langevin equation and thereby analytically generalize the equipartition theorem to a theory that we denote the non-Hermitian non-equipartition theory. We use the non-Hermitian non-equipartition theory to calculate the relevant statistics, which reveal that the averaged kinetic and potential energies are no longer equal to $k_B T/2$ and are not equipartitioned. As examples, we apply non-Hermitian non-equipartition theory to derive the connection between the non-Hermitian trapping force and particle statistics, whereby measurement of the latter can determine the former. Furthermore, we apply a non-Hermitian force to convert a saddle potential into a stable potential, leading to a different type of stable state.

Brownian motion is a fundamental type of thermal motion that is of paramount importance in various scientific and technological applications. The underlying mechanism of Brownian motion was explained by Einstein in 1905[1] and Smoluchowski in 1906[2], and its characteristic jittery movement results from the irregular bombardment of fluid molecules. This movement is random, complex, and unrepeatable, and thus Brownian motion can only be described by statistical theoretical treatments. Perhaps the most important and elegant result of studies on Brownian motion is the equipartition theorem (ET), which was first devised in 1843 for an equipartition of kinetic energies[3], and subsequently generalized[4], such that it became a cornerstone of classical statistical physics[5]. The ET states that at thermal equilibrium, every quadratic degree of freedom has an average energy of $k_B T/2$, where $k_B$ is the Boltzmann constant and $T$ is temperature. Furthermore, the ET plays a significant role in describing a broad range of physical scenarios, including the ideal gas law[5], the Dulong–Petit law for specific heat capacities of solids[6], Graham's law of effusion[7], the extreme relativistic ideal gas in astrophysics[5,8], and Johnson–Nyquist noise[9].

However, the ET is inadequate for certain contemporary problems, such as those associated with optical/acoustic trapping[10–18] and binding[19–32]. These involve a single particle or a system of particles bound at mechanical equilibrium by an external optical or acoustic wave, which causes the particle or system to be out of thermal equilibrium. The wave exchanges energy with the particle(s), leading to the generation of nonconservative forces that drive the particle or system into a non-Hermitian state[31]. This drastically alters the physical characteristics of a particle or system, and thus the ET fails to describe a particle or system whenever non-Hermitian manipulation is performed. External nonconservative forces compete with ambient damping, with the former pumping energy into particles and the latter removing energy from particles. Regarding optical manipulation, non-Hermitian forces are expected to play a vital role in optical trapping under vacuum (low damping) and a direct and significant role in optical trapping in air under atmospheric conditions (intermediate damping). Moreover, during optical trapping in water (heavy damping), non-Hermitian forces are expected to drive particles' characteristic vibrational modes into a non-orthogonal state that deviates from Hermitian physics.

In recent years, interest in non-Hermitian physics[33,34] has grown. This interest was initially sparked by studies in quantum mechanics[35–39] and then spread to a wide range of areas of physical science, including classical mechanics[40,41], optics[42–44], acoustics[45,46], metamaterials[47,48], electrical circuits[49–52], nuclear magnetic resonance[53], topological

[1]Department of Physics, Southern University of Science and Technology, Shenzhen, Guangdong 518055, China. [2]Department of Physics, The Hong Kong University of Science and Technology, Hong Kong, China. [3]Institute of Advanced Photonics, School of Physics, Harbin Institute of Technology, Harbin 150001, China. [4]These authors contributed equally: Xiao Li, Yongyin Cao. ✉e-mail: wuzh3@sustech.edu.cn

photonics[54–56], and optical manipulation[31,57–62]. We note that many of these topics are associated with classical physics. Here, we explore the application of non-Hermitian physics in Brownian dynamics. Specifically, one of our main themes is the application of non-Hermitian physics to optical trapping and binding, which are non-Hermitian systems. Non-Hermitian systems and optical trapping in vacuum or air (which are underdamped) hold significant and growing importance[29–31,58,59,61,62].

We use stochastic calculus to solve the Langevin stochastic differential equation with nonconservative trapping forces and thereby generalize the ET to the non-Hermitian non-equipartition (NHNE) theory (may be equivalently termed non-Hermitian non-equipartition theorem). We also use the Verlet algorithm[63] to validate the analytical stochastic calculations through numerical simulations. Our results significantly deviate from the $k_BT/2$ predicted by the ET, and the energies are no longer equipartitioned among the different degrees of freedom. Moreover, we observe qualitative discrepancies. For instance, a sufficiently large nonconservative force can destabilize an otherwise stable trap[14,31], whereas a particle in an originally unstable saddle potential can be stabilized by an appropriate nonconservative force. The NHNE theory is also capable of measuring forces other than conservative trapping forces[64], as it can measure non-Hermitian forces and the repulsive forces of a particle in a saddle potential. Additionally, we discuss the NHNE theory for $N > 1$ Brownian particles, which qualitatively captures accurate numerical results by applying an approximate analytical theory.

## Results
### Statement of the NHNE theory
Consider a spherical particle of mass $m$ that is immersed in a fluid and is confined within the vicinity of an equilibrium $\mathbf{r}' = (x', y', z')$ by a force field $\mathbf{F}(\mathbf{r}) = (F_x, F_y, F_z)$. The dynamics of the particle are governed by the Langevin stochastic differential equation[65]:

$$m\frac{d^2\mathbf{r}}{dt^2} = \mathbf{F}(\mathbf{r}) - \gamma\frac{d\mathbf{r}}{dt} + \mathbf{A}(t), \tag{1}$$

where $\mathbf{r} = (x, y, z)$ represents the position of the particle; $\gamma = 6\pi\eta a$ is the friction coefficient; $a$ is the particle radius; $\eta$ is viscosity; and $\mathbf{A}(t) = (A_x(t), A_y(t), A_z(t))$, which obeys the fluctuation–dissipation theorem, is the Gaussian-distributed random force due to Brownian fluctuations. $\mathbf{A}(t)$ has correlations of $\langle A_i(t)A_j(t')\rangle = 2\gamma k_B T\delta_{i,j}\delta(t-t')$, where $A_i(t)$ represents the $i$th component of $\mathbf{A}(t)$, $\delta_{i,j}$ is the Kronecker delta function, and $\delta(t)$ is the Dirac delta function.

Near the equilibrium position $\mathbf{r}'$, where $\mathbf{F}(\mathbf{r}') = \mathbf{0}$, $\mathbf{F}(\mathbf{r})$ in Eq. (1) may be sufficiently approximated by its linear term (as the zeroth-order term vanishes), as follows:

$$\mathbf{F}(\mathbf{r}) \approx \overleftrightarrow{\mathbf{K}} \cdot (\mathbf{r} - \mathbf{r}'), \tag{2}$$

where $\overleftrightarrow{\mathbf{K}}_{ij} = k_{ij} = \frac{\partial F_i}{\partial r_j}$ calculated at $\mathbf{r} = \mathbf{r}'$ is the force matrix[22,31], where $F_i$ and $r_i$ are the $i$th components of $\mathbf{F}(\mathbf{r})$ and $\mathbf{r}$, respectively. $\overleftrightarrow{\mathbf{K}}_{ij}$ and $F_i$ can be calculated numerically. For optical trapping, which we consider later, the force calculation is performed using Mie scattering theory and the Maxwell stress tensor[22,31,66]. It is equally valid to define the force matrix with an extra minus sign as $\mathbf{F}(\mathbf{r}) \approx -\overleftrightarrow{\mathbf{K}} \cdot (\mathbf{r} - \mathbf{r}')$, where a positive eigenvalue of $\overleftrightarrow{\mathbf{K}}$ indicates a stable mode. However, including an extra minus sign is a matter of preference and has no physical significance. Without loss of generality, one could take the equilibrium to be the origin, i.e., $\mathbf{r}' = \mathbf{0}$. By solving Eqs. (1) and (2) using stochastic calculus (Supplementary Notes 1, 2), we obtain

$$\begin{aligned}\langle r_i r_j\rangle &= \frac{2\gamma k_B T}{m^2}\sum_{n=1}^{3}\sum_{l=1}^{3}\sum_{m=1}^{3}[\overleftrightarrow{\Lambda}_{il}(\overleftrightarrow{\Lambda}^{-1})_{mn}][\overleftrightarrow{\Lambda}_{jm}(\overleftrightarrow{\Lambda}^{-1})_{ln}]\overleftrightarrow{\mathbf{M}}^\varphi_{ml},\\[4pt]\langle v_i v_j\rangle &= \frac{2\gamma k_B T}{m^2}\sum_{n=1}^{3}\sum_{l=1}^{3}\sum_{m=1}^{3}[\overleftrightarrow{\Lambda}_{il}(\overleftrightarrow{\Lambda}^{-1})_{mn}][\overleftrightarrow{\Lambda}_{jm}(\overleftrightarrow{\Lambda}^{-1})_{ln}]\overleftrightarrow{\mathbf{M}}^\phi_{ml},\end{aligned} \tag{3}$$

where $r_i (v_i)$ denotes the $i$th component of the displacement (velocity), the columns of $\overleftrightarrow{\Lambda}$ are the right eigenvectors for $-\overleftrightarrow{\mathbf{K}}/m$, and $\overleftrightarrow{\mathbf{M}}^\varphi_{ml}$ and $\overleftrightarrow{\mathbf{M}}^\phi_{ml}$ are given in the Methods. Equation (3) is one of the main results of this paper and generalizes the ET to non-Hermitian systems. A detailed derivation is presented in Supplementary Notes 1, 2. In general, unlike in the ET, the average energies in a non-Hermitian system associated with different degrees of freedom are not equipartitioned by Eq. (3). Hence, the generalization of the ET theory that it represents is referred to as the NHNE theory. We note that the NHNE theory, denoted by Eq. (3), can be applied to any force matrix $\overleftrightarrow{\mathbf{K}}$, including both Hermitian and non-Hermitian matrices, for a trapped Brownian particle.

### The NHNE theory for optical trapping
We provide a concrete example by considering optical trapping as a non-Hermitian system. In the case of a particle trapped by a wave, the non-Hermiticity of the force matrix results from light scattering[31,57,58]. We utilize Eq. (3) to investigate the specific scenario of a single particle trapped by light; the NHNE theory is also applicable to other mechanical systems, including acoustic trapping and binding systems. The motion of the trapped particle along the $z$-axis of a typical trapping beam is independent of the transverse motions whenever[31]

$$k_{xz} = \frac{\partial F_x}{\partial z} = k_{yz} = \frac{\partial F_y}{\partial z} = k_{zx} = \frac{\partial F_z}{\partial x} = k_{zy} = \frac{\partial F_z}{\partial y} = 0 \tag{4}$$

The conditions sufficient for Eq. (4) to hold include but are not limited to (i) the equilibrium state of the system exhibits mirror symmetry about the $z = z'$ plane (e.g., in the case of identical counter-propagating beams), (ii) the system exhibits mirror symmetries about the $x = x'$ and $y = y'$ planes at the equilibrium position (e.g., in the case of a linearly polarized Gaussian beam), or (iii) the system exhibits rotational symmetry (e.g., with a circularly polarized Gaussian beam). Here, $(x', y', z')$ is the equilibrium position of the particle. Equation (4) enables us to resolve the transverse motion by using the reduced two-dimensional (2D) non-Hermitian force matrix $\overleftrightarrow{\mathbf{K}}'_{2D} = \begin{bmatrix} k'_{xx} & k'_{xy} \\ k'_{yx} & k'_{yy} \end{bmatrix}$, where $\overleftrightarrow{\mathbf{K}}'^\dagger_{2D} \neq \overleftrightarrow{\mathbf{K}}'_{2D}$ as $k'_{xy} \neq k'_{yx}$ in general. By implementing a coordinate transformation that diagonalizes the symmetric part of $\overleftrightarrow{\mathbf{K}}'_{2D}$, we obtain, without loss of generality, $\overleftrightarrow{\mathbf{K}}_{2D} = \overleftrightarrow{\mathbf{R}}\overleftrightarrow{\mathbf{K}}'_{2D}\overleftrightarrow{\mathbf{R}}^{-1} = \begin{bmatrix} k_{xx} & g \\ -g & k_{yy} \end{bmatrix}$, where $\overleftrightarrow{\mathbf{R}}$ is a rotation matrix. Here, the existence of the anti-symmetric component $g$ in the force matrix is a consequence of non-Hermiticity. One can infer from the force field produced by $g$ that this nonconservative force revolves the particle around the mechanical equilibrium.

Then, Eq. (3) is simplified to

$$\begin{aligned}\frac{1}{2}m\langle v_x^2\rangle &= \left(\frac{1}{2} + \frac{2g^2}{\chi}\right)k_B T,\\[4pt]\frac{1}{2}m\langle v_y^2\rangle &= \left(\frac{1}{2} + \frac{2g^2}{\chi}\right)k_B T,\\[4pt]\frac{1}{2}m\langle v_x v_y\rangle &= \frac{-g(k_{xx}-k_{yy})}{\chi}k_B T,\\[4pt]-\frac{1}{2}k_{xx}\langle x^2\rangle &= \frac{k_{xx}}{\bar{k}}\left(\frac{1}{2} - k_{yy}\psi + \frac{2g^2}{\chi}\right)k_B T,\\[4pt]-\frac{1}{2}k_{yy}\langle y^2\rangle &= \frac{k_{yy}}{\bar{k}}\left(\frac{1}{2} + k_{xx}\psi + \frac{2g^2}{\chi}\right)k_B T,\\[4pt]-\frac{1}{2}g\langle xy\rangle &= -\frac{g^2}{\bar{k}}\left(\psi + \frac{(k_{xx}-k_{yy})}{\chi}\right)k_B T,\end{aligned} \tag{5}$$

where $\chi = -4g^2 + (k_{xx}-k_{yy})^2 - 2(k_{xx}+k_{yy})\gamma^2/m$, $\psi = (k_{xx}-k_{yy})/4(g^2+k_{xx}k_{yy})$, and $\bar{k} = (k_{xx}+k_{yy})/2$. Equation (5) represents the 2D

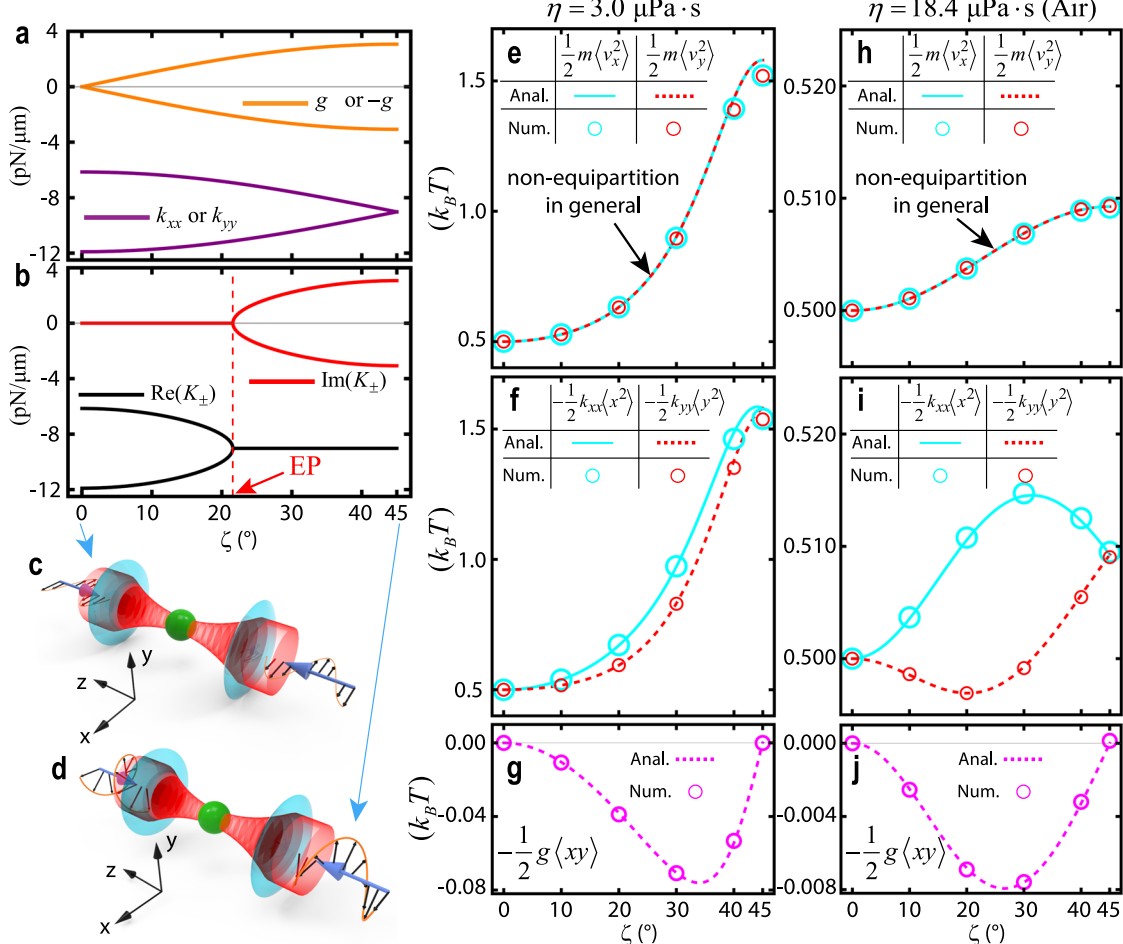

**Fig. 1 | Non-Hermitian non-equipartitioning in optical trapping. a** Components of the force matrix $\overset{\leftrightarrow}{\mathbf{K}}_{2D} = \begin{bmatrix} k_{xx} & g \\ -g & k_{yy} \end{bmatrix}$ and **b** its eigenvalues for an optically trapped dielectric Brownian particle with refractive index $n = 1.57$ and radius $a = 0.5\ \mu m$ in a low vacuum or in air, as the polarization ($\hat{\mathbf{p}} = \hat{\mathbf{x}}\cos\zeta + i\hat{\mathbf{y}}\sin\zeta$) of the incident beams (with wavelength $\lambda = 1.064\ \mu m$) varies from linear ($\zeta = 0°$, panel **c**) to circular ($\zeta = 45°$, panel **d**). The red dashed line in panel **b** marks the exceptional point (EP). The focused beam has a numerical aperture of 0.9 and a filling factor of 1.0, and the beam power for each beam is normalized to 1.0 mW. For different viscosities, $\eta = 3.0\ \mu Pa\cdot s$ (**e**–**g**) and $\eta = 18.4\ \mu Pa\cdot s$ (**h**–**j**), average energies $\frac{1}{2}m\langle v_x^2\rangle$ and $\frac{1}{2}m\langle v_y^2\rangle$, $-\frac{1}{2}k_{xx}\langle x^2\rangle$ and $-\frac{1}{2}k_{yy}\langle y^2\rangle$, and $-\frac{1}{2}g\langle xy\rangle$ are presented as lines (Anal.: analytical values) and circles (Num.: numerical values).

NHNE theory, which is fundamentally distinct from the ET. Here, we note that the temperature $T$ refers to the ambient temperature. It is unchanged for transparent particles since the work done by the non-Hermitian force is on the order of $k_B T$ (comparable with the energy of a single molecule in the ambient), thus completely negligible for the entire ambient. When $g = 0$, $\overset{\leftrightarrow}{\mathbf{K}}_{2D}$ is Hermitian, and Eq. (5) reduces to the ET, where $\frac{1}{2}m\langle v_x^2\rangle = \frac{1}{2}m\langle v_y^2\rangle = -\frac{1}{2}k_{xx}\langle x^2\rangle = -\frac{1}{2}k_{yy}\langle y^2\rangle = \frac{1}{2}k_B T$. However, when $g \neq 0$, Eq. (5) can significantly deviate from the ET. The equality of kinetic energy in Eq. (5) results from our aiming to simplify the equation by selecting a specific coordinate system (where $\overset{\leftrightarrow}{\mathbf{K}}_{2D} = \begin{bmatrix} k_{xx} & g \\ -g & k_{yy} \end{bmatrix}$). The general form for arbitrary $\overset{\leftrightarrow}{\mathbf{K}}_{2D}$ is available in Supplementary Note 1. Generally, the kinetic energies along two orthogonal transverse directions are not equal, as explained in Supplementary Note 3. If the non-Hermitian force matrix is scaled as $P\overset{\leftrightarrow}{\mathbf{K}}_{2D}$, the averaged quantities in Eq. (5) are nonlinear with respect to the scalar $P$, as altering $P$ changes the ratio between the trapping forces and the random forces. This also applies to Eq. (3). In optical and acoustic trapping, $P$ is proportional to the incident power, as detailed in Supplementary Note 6.

We now apply Eq. (5) to investigate a Brownian particle (with refractive index $n = 1.57$ and radius $a = 0.5\ \mu m$) that is illuminated and trapped by two counter-propagating Gaussian beams that each have an input power of 1.0 mW. Such a system exhibits a non-Hermitian force matrix that is tunable by varying the incident polarization $\hat{\mathbf{p}} = \hat{\mathbf{x}}\cos(\zeta) + i\hat{\mathbf{y}}\sin(\zeta)$. A similar system was examined in ref. 31, but without considering Brownian motion. The values of the components of $\overset{\leftrightarrow}{\mathbf{K}}_{2D}$ versus the incident polarizations are illustrated in Fig. 1a, while the corresponding eigenvalues,

$$K_\pm = \bar{k} \pm \sqrt{(k_{xx} - k_{yy})^2/4 - g^2},\qquad(6)$$

are shown in Fig. 1b, where $k_{xx}$ and $k_{yy}$ represent the restoring force constants in the $x$ and $y$ directions, respectively. The parameter $g$ represents the torque driven by the orbital angular momentum of the elliptically or circularly polarized Gaussian beam, which is created by the spin–orbit angular momentum conversion during focusing by the objective lens[67]. As $\zeta$ is continuously tuned from 0° to 45°, the polarization gradually varies from linear (Fig. 1c) to elliptical and then to circular (Fig. 1d). With linear polarization ($\zeta = 0°$), $g = 0$ due to reflection symmetries on the $xz$ and $yz$ planes, and thus the system is effectively Hermitian. In addition, $k_{xx} \neq k_{yy}$ due to polarization aberrations[68]. As $\zeta$ increases to (for example) 10°, the left circular polarization becomes stronger than the right circular polarization, which breaks the balance of the orbital angular momentum originally

presented in the linear polarization and generates a finite $g$. Moreover, the focused spot approaches cylindrically symmetric, and thus the difference between $k_{xx}$ and $k_{yy}$ decreases. At $\zeta = 21.6°$, which is indicated by the red dashed line in Fig. 1b, an exceptional point (EP) emerges, where the real parts of the eigenvalues merge and their imaginary parts split. The EP arises due to a switch in the sign of the quantity under the square root in Eq. (6), which causes the originally real eigenvalues to become complex. This is expected, as $(k_{xx} - k_{yy})^2$ ultimately reaches zero (at $\zeta = 45°$) due to rotational symmetry, while $g$ increases monotonically from zero. As a result, the existence of an EP at $g^2 = (k_{xx} - k_{yy})^2$ is inevitable, regardless of the specifics of a system.

The values of $K_\pm$ in Eq. (6) are real and negative (indicating stability) to the left of the EP, whereas they become complex (indicating instability if ambient damping is insufficient)[31] to the right of the EP. Figure 1e shows the kinetic energies of $\frac{1}{2}m\langle v_x^2\rangle$ and $\frac{1}{2}m\langle v_y^2\rangle$ plotted for a viscosity of $\eta = 3.0\,\mu\mathrm{Pa\cdot s}$, whereas Fig. 1f shows the quantities $-\frac{1}{2}k_{xx}\langle x^2\rangle$ and $-\frac{1}{2}k_{yy}\langle y^2\rangle$. Crucially, the last two quantities, referred to as potential energies, only represent the conservative interaction, but not the entire non-Hermitian interactions. The latter cannot be expressed as the gradient of a potential. According to the ET, all four quantities should equal $k_\mathrm{B}T/2$. However, they deviate from $k_\mathrm{B}T/2$, except when the system is effectively Hermitian under linear polarization ($\zeta = 0°$). Figure 1g also displays $-\frac{1}{2}g\langle xy\rangle$, which has non-zero values, indicating the presence of non-Hermitian couplings between the $x$ and $y$ motions. Due to the axial symmetry, $-\frac{1}{2}g\langle xy\rangle = 0$ under circular polarization ($\zeta = 45°$), despite non-Hermitian coupling still being present. We compare the results obtained using the NHNE theory (Eq. (5)) with those generated by a numerical Verlet simulation (marked by circles in Fig. 1e–g). The Verlet simulation utilizes the "exact" optical force field computed from Mie scattering theory[66], instead of the linearized force based on the force matrix. In the Mie scattering theory, the expansion series are truncated at $L_\mathrm{max} = ka + 4(ka)^{1/3} + 2$[69,70], with $k = \frac{2\pi}{\lambda}$ being the wavenumber. In some cases, we verified the convergence of our calculations by comparing the calculation truncated at $L_\mathrm{max}$ with those truncated at $L_\mathrm{max} + 5$. Further details on the Verlet simulation can be found in the Methods. The results show remarkable agreement, with small deviations found only near areas of circular polarization. This is because circular polarization has the largest $g$, which leads to the fastest rate of energy pumping, allowing a particle to move further away from the origin than under other types of polarization, such that the linear approximation in Eq. (2) becomes less accurate. The agreement with the "exact" force calculation demonstrates the sufficiency of the linear approximation used to derive Eq. (5).

Figure 1h–j show the same settings as Fig. 1e–g, except that the former has a viscosity $\eta = 18.6\,\mu\mathrm{Pa\cdot s}$ (air), whereas the latter has $\eta = 3.0\,\mu\mathrm{Pa\cdot s}$. As the damping increases, the maximum deviations from the ET decrease from ~200% to ~20%. This indicates that although the non-Hermiticity is significantly suppressed, it cannot be ignored, even for optical trapping in air. A large dissipation implies that the energy the particle receives from light is dissipated quickly, preventing its accumulation. Regardless of the level of damping, the presence of non-Hermitian forces ensures that the vibrational eigenmodes are always non-orthogonal, indicating that the non-Hermiticity of the force matrix cannot be ignored in any case. In all cases, the results from the NHNE theory remain accurate. The non-Hermiticity of optical trapping also depends strongly on the particle radius ($a$), as detailed in Supplementary Note 8.

Interestingly, $-\frac{1}{2}k_{xx}\langle x^2\rangle$ and $-\frac{1}{2}k_{yy}\langle y^2\rangle$ in Fig. 1f, i are not the same. In fact, in Fig. 1i, $-\frac{1}{2}k_{yy}\langle y^2\rangle$ can even be less than $k_\mathrm{B}T/2$. This observation can be explained by the virial theorem[71], $\frac{1}{2}m\langle v_i^2\rangle = \sum_j -\frac{1}{2}\ddot{\mathbf{K}}_{ij}\langle r_i r_j\rangle$, which remains applicable even in the non-Hermitian case (Supplementary Note 3). In addition, according to the virial theorem, one can obtain $\frac{1}{2}m\langle v_x^2\rangle = -\frac{1}{2}k_{xx}\langle x^2\rangle - \frac{1}{2}g\langle xy\rangle$ and $\frac{1}{2}m\langle v_y^2\rangle = -\frac{1}{2}k_{yy}\langle y^2\rangle + \frac{1}{2}g\langle xy\rangle$.

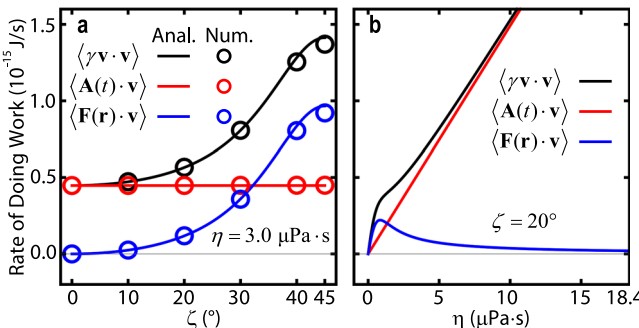

**Fig. 2 | Power delivered by different force components.** Power delivered by the damping force $\langle\gamma\mathbf{v}\cdot\mathbf{v}\rangle$ (black), the fluctuational force $\langle\mathbf{A}(t)\cdot\mathbf{v}\rangle$ (red), and the optical force $\langle\mathbf{F}(\mathbf{r})\cdot\mathbf{v}\rangle$ (blue) versus **a** the polarizations ($\zeta$) and **b** viscosities ($\eta$) in the optical trapping of a Brownian particle (identical to that described in Fig. 1e–g) is presented. The analytical (Anal.) and numerical (Num.) values are denoted by lines and circles, respectively.

Figure 2 shows the average rate at which work is done on the particle by the dissipative force $-\langle\gamma\mathbf{v}\cdot\mathbf{v}\rangle$, random force $\langle\mathbf{A}(t)\cdot\mathbf{v}\rangle$, and optical force $\langle\mathbf{F}(\mathbf{r})\cdot\mathbf{v}\rangle$, calculated analytically (lines) and numerically (circles), as a function of polarizations (Fig. 2a) and viscosities (Fig. 2b). As required by energy conservation, we have $-\langle\gamma\mathbf{v}\cdot\mathbf{v}\rangle + \langle\mathbf{A}(t)\cdot\mathbf{v}\rangle + \langle\mathbf{F}(\mathbf{r})\cdot\mathbf{v}\rangle = 0$, as detailed in Supplementary Note 4. As $\mathbf{A}(t)$ is a random force that does not depend on $\mathbf{v}$, its power is expected to be the same for a particle moving freely and a particle moving under a force field, resulting in a constant $\langle\mathbf{A}(t)\cdot\mathbf{v}\rangle = \frac{2\gamma k_\mathrm{B}T}{m}$ in Fig. 2a. Moreover, in Fig. 2a, $\langle\mathbf{F}(\mathbf{r})\cdot\mathbf{v}\rangle$ increases with the non-Hermiticity of the force matrix as $\zeta$ increases, which accounts for the increased energies in the NHNE theory results shown in Fig. 1e–j. As $\eta$ increases, the role of the optical force initially increases and then decreases to zero, because at a large $\eta$, work is done at a high rate by the Brownian fluctuation $\langle\mathbf{A}(t)\cdot\mathbf{v}\rangle$ and the corresponding motion is heavily damped. Surprisingly, the optical force does no work in a perfect vacuum, but it does work when fluctuation and damping are present; this highlights an unexpected role of Brownian motion in light-driven machines.

### The NHNE theory for a saddle potential with a nonconservative force field
A non-Hermitian force matrix consists of a potential energy term (the symmetric part of $\ddot{\mathbf{K}}_{2\mathrm{D}}$) and a nonconservative term (the antisymmetric part of $\ddot{\mathbf{K}}_{2\mathrm{D}}$). A saddle potential traps a particle in one direction (when $k_{xx} < 0$) and repels the particle in the other direction (when $k_{yy} > 0$), making it unstable. Counter-intuitively, a non-Hermitian force can stabilize a particle in a saddle potential if the trace of $\ddot{\mathbf{K}}_{2\mathrm{D}}$ is negative. As the polarization ($\hat{\mathbf{p}} = \hat{\mathbf{x}}\cos(\zeta) + i\hat{\mathbf{y}}\sin(\zeta)$) varies from linear ($\zeta = 0°$, Fig. 3c) to circular ($\zeta = 45°$, Fig. 3d), the matrix elements of $\ddot{\mathbf{K}}_{2\mathrm{D}}$, namely $k_{xx}$, $k_{yy}$, $g$, and $-g$, are illustrated with lines in Fig. 3a for optical trapping of a hollow sphere (with refractive index $n = 1.57$, inner radius 0.49 $\mu$m, and outer radius $a = 0.70$ $\mu$m), where the saddle potential (defined by $k_{xx}\cdot k_{yy} < 0$) occurs on the left-hand side of the black dashed line. Figure 3b plots $\mathrm{Re}(K_\pm)$ and $\mathrm{Im}(K_\pm)$ versus $\zeta$, wherein the EP is marked with a red dashed line, and the neutral point (NP) with $K_+ = 0$ and $K_- < 0$ is marked with a blue dashed line. Coincidentally, the black dashed line (Fig. 3a) and the EP are very close. On the left-hand side of the NP, the positive $K_+$ mode repels the particle, regardless of the ambient damping level, whereas on the right-hand side of the EP, the complex modes $K_\pm$ destabilize the particle when the background damping is not sufficiently large. Unexpectedly, between the NP and the EP, the particle is always stable. Similar to Fig. 1, we depict the NHNE in Fig. 3e–j) with viscosity $\eta = 3.0\,\mu\mathrm{Pa\cdot s}$ ($\eta = 18.4\,\mu\mathrm{Pa\cdot s}$) versus polarizations. At the NP, $-\frac{1}{2}k_{xx}\langle x^2\rangle$, $-\frac{1}{2}k_{yy}\langle y^2\rangle$, and $-\frac{1}{2}g\langle xy\rangle$ diverge because there is no confinement by optical force along the neutral

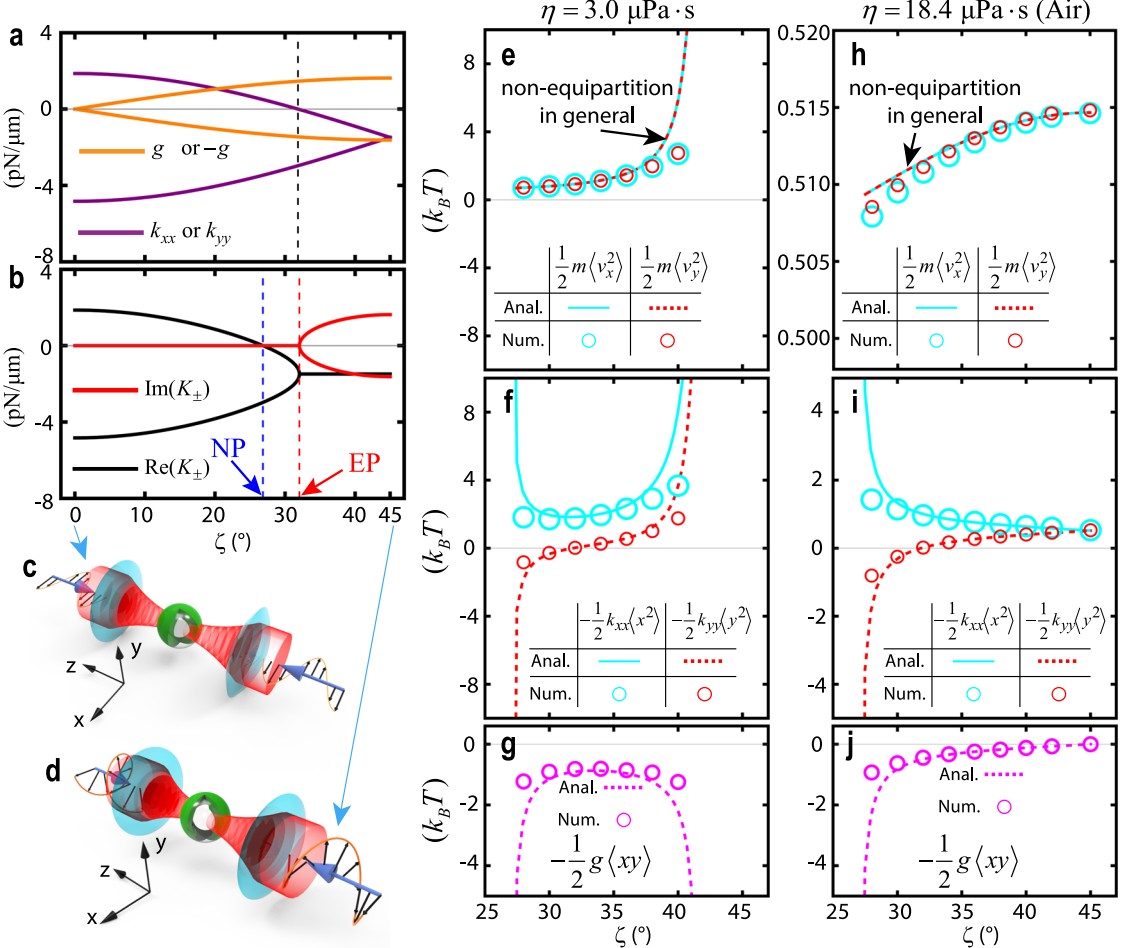

**Fig. 3 | Non-Hermitian non-equipartitioning in optical trapping of a saddle potential.** Components (**a**) and eigenvalues (**b**) of the force matrix

$\overleftrightarrow{\mathbf{K}}_{2D} = \begin{bmatrix} k_{xx} & g \\ -g & k_{yy} \end{bmatrix}$ for an optically trapped dielectric hollow particle with refractive index $n = 1.57$, inner radius 0.49 μm, and outer radius $a = 0.70$ μm, in a low vacuum or in air, as the polarization ($\hat{\mathbf{p}} = \hat{\mathbf{x}}\cos\zeta + i\hat{\mathbf{y}}\sin\zeta$) of the incident beams (with wavelength $\lambda = 1.064$ μm) varies from linear ($\zeta = 0°$, **c**) to circular ($\zeta = 45°$, **d**). The left-hand side of the black dashed line in **a** corresponds to the saddle potential,

where $k_{xx} \cdot k_{yy} < 0$. The blue and red dashed lines in **b** indicate the neutral point (NP), where one of the eigenvalues $K_{\pm}$ is 0, and the exceptional point (EP), respectively. The focused beam has a numerical aperture of 0.9 and a filling factor of 1.0, and the power of each beam is normalized to 1.0 mW. For different viscosities, $\eta = 3.0$ μPa·s (**e**–**g**) and $\eta = 18.4$ μPa·s (**h**–**j**), average energies $\frac{1}{2}m\langle v_x^2\rangle$ and $\frac{1}{2}m\langle v_y^2\rangle$, $-\frac{1}{2}k_{xx}\langle x^2\rangle$ and $-\frac{1}{2}k_{yy}\langle y^2\rangle$, and $-\frac{1}{2}g\langle xy\rangle$ are presented as lines (Anal.: analytical values) and circles (Num.: numerical values).

direction. However, the kinetic energies remain finite. For $\eta = 3.0$ μPa·s, $\langle v_x^2\rangle$, $\langle v_y^2\rangle$, $\langle x^2\rangle$, $\langle y^2\rangle$, and $\langle xy\rangle$ diverge beyond the polarization characterized by $\zeta = 41.7°$, i.e., after the EP, because $K_{\pm}$ are complex and the background damping is insufficient, making the trapping unstable. Figure 3e–j present the analytical (lines) and numerical (circles) calculations. Inconsistencies are only observed when the particle is far away from the equilibrium position. As $k_{yy}$ varies from positive to negative due to the increase in $\zeta$, $-\frac{1}{2}k_{yy}\langle y^2\rangle$ also varies from negative to positive (Fig. 3f, i), i.e., from repulsion to trapping. This is a unique phenomenon for a saddle potential with a non-Hermitian force field. As an additional example of saddle potential, we also consider a uniform layer of dielectric coated on a gold sphere for the purpose of optical trapping, as illustrated in Supplementary Note 9.

## Trajectories for optical trapping
The trajectories of particles trapped by non-Hermitian forces and subject to Brownian motion are depicted in Fig. 4a, b (trapping potential) and Fig. 4c, d (saddle potential), with their initial positions marked by black dots. For the trapping potential at $\eta = 1.0$ μPa·s (Fig. 4a), the viscosity is too low to confine the particle after the EP. However, when the viscosity is increased to $\eta = 3.0$ μPa·s, as shown in

Fig. 4b, the Brownian particle is stable at all of the polarizations. We can still observe the expansion of the trajectories as $\zeta$ increases[58], which is a result of the increasing $\langle x^2\rangle$ and $\langle y^2\rangle$ shown in Fig. 1f. In addition, the trapped particle exhibits Brownian fluctuations near the equilibrium, and its orientation varies with $\zeta$ (see Supplementary Note 5).

For the saddle potential, the particle escapes before the NP ($\zeta < 26.9°$), due to the repulsive force (Fig. 4c, d), and after $\zeta = 32.6°$ ($\zeta = 41.7°$), when the viscosity is as low as $\eta = 1.0$ μPa·s ($\eta = 3.0$ μPa·s), as shown in Fig. 4c (Fig. 4d). On the NP, the Brownian particle is trapped in only one direction, whereas the fluctuation force repels the particle far away in another direction. Between the NP and the EP ($26.9° < \zeta < 32.0°$), the Brownian particle is trapped stably, even though the optical force repels the particle in one direction. The videos for the three-dimensional trajectories for each case depicted in Fig. 4a–d are available in Supplementary Movies 1–4, and detailed phase diagrams for both trapping and saddle potentials can be found in Supplementary Note 6.

## The NHNE theory for multiple particles
Compared with the NHNE theory for a single particle, the NHNE theory for multiple particles is significantly more complex due to hydrodynamic interactions between particles[72]. Currently, there is no exact

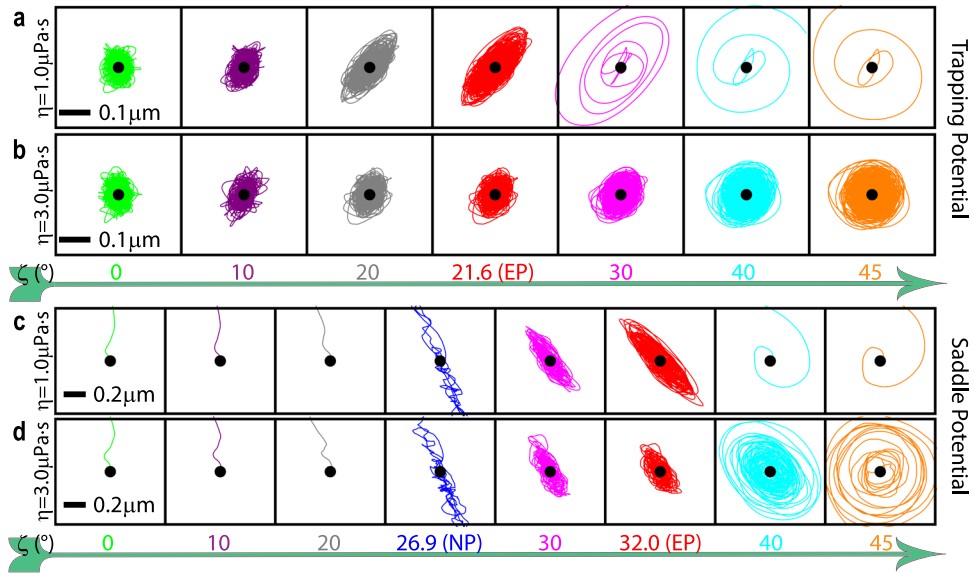

**Fig. 4 | Trajectories for an optically manipulated Brownian particle.** Numerically calculated trajectories for viscosities $\eta = 1.0$ µPa·s (**a**, **c**) and $\eta = 3.0$ µPa·s (**b**, **d**) with respect to various polarizations ($\zeta$) for a Brownian particle being optically manipulated in a trapping potential (**a**, **b**, corresponding to Fig. 1) or a saddle potential (**c**, **d**, corresponding to Fig. 3) are presented. The black dots indicate the coordinate origins and the starting positions of the trajectories, and the initial velocity of the particle is 0.

analytical solution available, but one can use the Verlet algorithm to solve the problem numerically, as discussed in Supplementary Note 7. Furthermore, an approximate solution can be derived if the hydrodynamic interactions are neglected (see Supplementary Note 7):

$$\langle r_i r_j \rangle = \frac{2\gamma k_B T}{m^2} \sum_{n=1}^{3N} \sum_{l=1}^{3N} \sum_{m=1}^{3N} [\overleftrightarrow{\mathbf{\Lambda}}_{il}(\overleftrightarrow{\mathbf{\Lambda}}^{-1})_{mn}][\overleftrightarrow{\mathbf{\Lambda}}_{jm}(\overleftrightarrow{\mathbf{\Lambda}}^{-1})_{ln}]\overleftrightarrow{\mathbf{M}}^{\varphi}_{ml},$$

$$\langle v_i v_j \rangle = \frac{2\gamma k_B T}{m^2} \sum_{n=1}^{3N} \sum_{l=1}^{3N} \sum_{m=1}^{3N} [\overleftrightarrow{\mathbf{\Lambda}}_{il}(\overleftrightarrow{\mathbf{\Lambda}}^{-1})_{mn}][\overleftrightarrow{\mathbf{\Lambda}}_{jm}(\overleftrightarrow{\mathbf{\Lambda}}^{-1})_{ln}]\overleftrightarrow{\mathbf{M}}^{\phi}_{ml}.$$

(7)

The approximate analytical solutions semi-qualitatively agree with the numerical results.

We consider a linear chain of spheres with $N = 2$ (Fig. 5a) that are optically trapped and bounded in a low vacuum by two coherent linearly polarized plane waves ($\eta = 1.0$ µPa·s) propagating along the chain axis. The two plane waves have different intensities $I_1$ and $I_2$ ($I_1 = 9I_2$), respectively, creating an unbalanced propagation that favors the non-Hermitian force fields[73,74]. Each sphere has a radius of $a = 0.2$ µm and a refractive index of $n$. We search for an equilibrium configuration for the pair of spheres and consider their motion along the $z$-axis. The red lines in Fig. 5b (with separation $D \approx \lambda$ and refractive index $n = 1.1$) and Fig. 5c (with $D \approx 4\lambda$ and $n = 1.2$) plot the averaged kinetic energies $\frac{1}{2}m\langle v_{z,i}^2 \rangle$, which are calculated using Eq. (7) for each sphere $i$ versus the averaged intensity $\bar{I}_0 = \frac{1}{2}(I_1 + I_2)$. Here, $\overleftrightarrow{\mathbf{K}} = \bar{I}_0 \overleftrightarrow{\mathbf{K}}_0$, where $\overleftrightarrow{\mathbf{K}}_0$ is the force matrix at $\bar{I}_0 = 1.0$ W/m². As the components of the non-Hermitian force matrix increase with $\bar{I}_0$, the kinetic energies increasingly deviate from $k_B T/2$ and are no longer equipartitioned. We also conduct numerical simulations using Verlet algorithms that take the hydrodynamic interactions into account. The results are depicted as blue dots in Fig. 5b (for a small $D$) and Fig. 5c (for a large $D$). The hydrodynamic interactions have a significant impact on the partitioned energies when $D$ is small, i.e., the blue dots deviate from the red line, as shown in Fig. 5b. However, the deviation diminishes when $D$ is large, as the hydrodynamic interaction is weakened, as shown in Fig. 5c. In sum, the approximate multiple particle NHNE theory (Eq. (7)), from which hydrodynamic interactions are excluded, can semi-qualitatively predict the averaged energies for each degree of freedom, especially when $D$ is large.

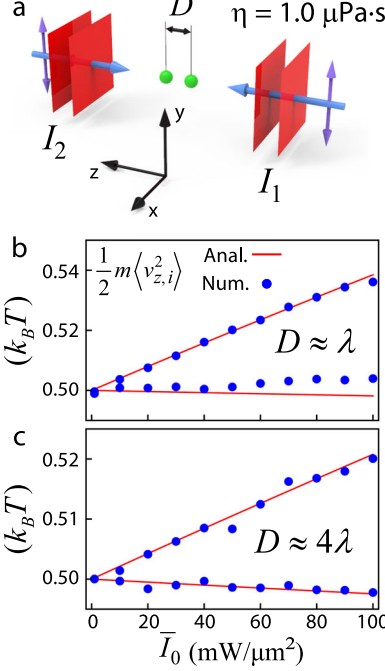

**Fig. 5 | Non-Hermitian non-equipartitioning in the optical binding of two spheres. a** The two spheres (colored green), each with a radius of $a = 0.2$ µm and a refractive index of $n$, are optically bounded by two linearly polarized counter-propagating plane waves (with wavelength $\lambda = 1.064$ µm) in a low vacuum ($\eta = 1.0$ µPa·s). The two plane waves have different intensities, with $I_1 = 9I_2$, and $\bar{I}_0 = \frac{1}{2}(I_1 + I_2)$. The average kinetic energies $\frac{1}{2}m\langle v_{z,i}^2 \rangle$ for the particles (indexed by $i$, ranging from 1 to 2) versus $\bar{I}_0$ are presented as red lines (analytical (Anal.) results excluding hydrodynamic interactions) and blue dots (numerical (Num.) results including hydrodynamic interactions) for different separations ($D$) and different values of $n$, namely **b** $D \approx \lambda$ and $n = 1.1$, and **c** $D \approx 4\lambda$ and $n = 1.2$. This analysis focuses on motion along the $z$-axis.

## Discussion

In this article, stochastic calculus is applied to generalize the ET to deal with non-Hermitian trapping and binding forces. This

generalized theory, denoted by the NHNE theory, enables the calculation of the average energies of a single particle or a group of trapped or bounded particles, even when the force matrix of a system is non-Hermitian. By "generalize", we meant to extend the original ET to address non-Hermitian problems. We note that the NHNE theory reveals the breaking of universality in the original ET by non-Hermiticity, in the sense that the average energies associated with each degree of freedom are no longer equal and depends on the details of the system. This is a development in the study of Brownian motion and has far-reaching implications for a variety of problems associated with modern technology, including those associated with optical/acoustic trapping and binding, and with other open mechanical systems.

To provide a concrete illustration of the NHNE theory, we focus on optical trapping, which is one example of a non-Hermitian trapping system. The NHNE theory is applied to analyze both the trapping potential (Fig. 1a) and the saddle potential (Fig. 3a). We propose that by experimentally measuring $\langle x^2 \rangle$, $\langle y^2 \rangle$, $\langle xy \rangle$, $\langle v_x^2 \rangle$, $\langle v_y^2 \rangle$, and $\langle v_x v_y \rangle$, Eq. (5) can be used to determine both the Hermitian ($k_{xx}$ and $k_{yy}$) and non-Hermitian ($g$) force constants. To our knowledge, such an approach is previously limited by the availability of pertinent theories, as a result, they can only be used to perform such indirect measurement of force constants in a heavily damped environment[57,75]. Thus, the NHNE theory provides a method to directly measure force constants under arbitrary levels of damping, including in vacuum trapping applications. When there is a large damping, some predictions by our theory (such as the average energies) can be very similar to what conventional ET predicts. It might seem like damping is getting rid of non-Hermiticity, but that is not the complete physical picture. For example, the vibrational eigenmodes remain non-orthogonal, irrespective of the damping. Moreover, we make the surprising finding that non-Hermitian forces can stabilize a particle in a saddle potential. Repulsive forces at the microscopic scale can be difficult to measure, due to the absence of a stable equilibrium. We propose that a microparticle located near a saddle potential can be stabilized by non-Hermitian forces. Furthermore, the repulsive force constant in a saddle potential can be determined using the NHNE theory.

We note that the NHNE theory is of relevance to the study of non-reciprocal interaction and active matters[76–80]. These systems are typically complex, involving intricate geometries, and a variety of non-reciprocal and non-Hermitian interactions. Our theory may offer some insights and approximate predictions into these problems.

## Methods
### Expressions of $\overleftrightarrow{\mathbf{M}}_{ml}^{\varphi}$ and $\overleftrightarrow{\mathbf{M}}_{ml}^{\phi}$

In Eq. (3),

$$
\begin{cases}
\overleftrightarrow{\mathbf{M}}_{ml}^{\varphi} = -\frac{\mu_m^+ + \mu_m^- + \mu_l^+ + \mu_l^-}{(\mu_m^+ + \mu_l^+)(\mu_m^- + \mu_l^+)(\mu_m^+ + \mu_l^-)(\mu_m^- + \mu_l^-)}, \\
\overleftrightarrow{\mathbf{M}}_{ml}^{\phi} = -\frac{\mu_m^- \mu_l^+ (\mu_m^- + \mu_l^+) + \mu_m^+ \mu_l^- (\mu_m^- + \mu_l^+)}{(\mu_m^+ + \mu_l^+)(\mu_m^- + \mu_l^+)(\mu_m^+ + \mu_l^-)(\mu_m^- + \mu_l^-)},
\end{cases} \quad (8)
$$

where

$$
\mu_i^{\pm} = \frac{1}{2}\left( -\frac{\gamma}{m} \pm \sqrt{\left(\frac{\gamma}{m}\right)^2 - 4\omega_i^2} \right), \quad (9)
$$

and $\omega_i = \sqrt{K_i}$ with $K_i$ being the $i$th eigenvalue of $-\overleftrightarrow{\mathbf{K}}/m$.

### Verlet algorithm for Langevin dynamics simulations
We use the Verlet algorithm[63] to solve the stochastic Langevin differential equation in Eq. (1) numerically. By definition, the fluctuating force $\mathbf{A}(t) = (A_x(t), A_y(t), A_z(t))$, which satisfies $\langle A_i(t) A_j(t') \rangle = 2\gamma k_\mathrm{B} T \delta_{i,j} \delta(t - t')$, is independent of the particle velocity $\mathbf{v}(t)$, and one can assume that $\mathbf{A}(t)$ is a constant force during each time step $h$. We

denote the fluctuating force during the time interval $[t, t+h]$ ($[t, t-h]$) as $\mathbf{A}_+$ ($\mathbf{A}_-$). The particle positions at time $t$, $t-h$, and $t+h$ are related by

$$
\mathbf{r}(t+h)(m + \gamma h/2) + \mathbf{r}(t-h)(m - \gamma h/2) = 2\mathbf{r}(t)m + h^2(\mathbf{F}(\mathbf{r}(t)) + \mathbf{A}_+/2 + \mathbf{A}_-/2), \quad (10)
$$

while for the particle velocities,

$$
\mathbf{v}(t+h/2)(m + \gamma h/2) = \mathbf{v}(t-h/2)(m - \gamma h/2) + h(\mathbf{F}(\mathbf{r}(t)) + \mathbf{A}_+/2 + \mathbf{A}_-/2). \quad (11)
$$

Here, $\mathbf{F}(\mathbf{r}(t))$ denotes the external force (e.g., optical force) exerted on the particles located at $\mathbf{r}(t)$. The Verlet results presented in the main text and Supplementary Information are obtained using a time step $h = 10^{-8}$ s. The average quantities are based on the average of $10^{10}$ positions or velocities.

### Reporting summary
Further information on research design is available in the Nature Portfolio Reporting Summary linked to this article.

### Data availability
The data that support the findings of this study are available at https://figshare.com/articles/dataset/Data_Availability_for_Nature_Communications_Non-Hermitian_Non-Equipartition_Theory_for_Trapped_Particles/25196924.

### Code availability
The codes used for the Brownian motion of an optically trapped particle are available at https://figshare.com/articles/dataset/Code_Availability_for_Nature_Communications_Non-Hermitian_Non-Equipartition_Theory_for_Trapped_Particles/25197044.

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

## Acknowledgements

The authors would like to thank Prof. C. T. Chan and Prof. Zhao-Qing Zhang for the stimulating discussions. This work is supported by the National Natural Science Foundation of China (no. 12074169), the Stable Support Plan Program of Shenzhen Natural Science Fund (no. 20200925152152003), and the Guangdong Province Talent Recruitment Program (no. 2021QN02C103). Y.C. acknowledges the support from the National Natural Science Foundation of China (No. 12274105), Heilongjiang Natural Science Funds for Distinguished Young Scholar (No. JQ2022A001), and Fundamental Research Funds for the Central Universities (HIT.OCEF.2021020).

## Author contributions

All authors discussed the results thoroughly. Most simulations were performed by X.L. and some are performed by Y.C. Y.C. derived the initial form of the NHNE theory, and X.L. perfected it and generalized it to higher dimensions. The physics and data were analyzed by J.N. and X.L. J.N. initiated and oversaw the project. X.L. and J.N. wrote the paper with input from all authors. X.L. and Y.C. contributed equally to this work.

## Competing interests

The authors declare no competing interests.
