## [Peer Review File · Nature Communications]

Reviewers' Comments:

Reviewer #1:

Remarks to the Author:

This work explores the generalized equipartition theorem of Brownian particles trapped by a non-Hermitian force field. It is a beautiful interdisciplinary study that combines non-Hermitian physics, Brownian motion, the Equipartition Theorem, and optical trapping. The authors demonstrate that in non-Hermitian force fields, the averaged energies are not evenly distributed and deviate from the expected value of $kBT/2$ per degree of freedom. The authors derive analytical formulas to describe the behavior of particles trapped by non-Hermitian force fields, which align with the Equipartition Theorem for Hermitian force fields. Additionally, the authors show that these formulas, known as the "Non-Hermitian Non-Equipartition Theory," can measure both trapping and repulsing force fields, regardless of their Hermitian or non-Hermitian nature. Numerical simulations provide nice verifications to the analytical derivations.

In my view, this manuscript addresses an important and longstanding problem from a modern perspective of non-Hermitian physics. It is likely to be of interest to both theorists and experimentalists. The presented analytical formulas not only provide insights into the behavior of particles in non-Hermitian force fields, but also offer an extension to the Equipartition Theorem. The significance of this study is further enhanced by the prevalence of non-Hermitian systems in the real world. Overall, this is an excellent piece of work with broad appeal. Nature Communications is a suitable venue. However, I do have some concerns that I would like the authors to address.

1. It was stated multiple times in the manuscript that the theory is applicable to acoustic trapping. However, the range of particle size where acoustic trapping is applicable spanning from microscopic to macroscopic. Please be more specific on the range where the theory is applicable.
2. Apparently, the non-Hermitian physics is less important when ambient damping increases, for example, from low vacuum to air. Does this suggest that ambient damping counteracts the influence of non-Hermitian force fields? Is it correct to say that the theory is mainly applicable in environments with minimal ambient damping?
3. Fabricating a hollow micro-sphere can be challenging. Could the author provide an alternative example that demonstrates the realization of the saddle potential? Please clarify whether the non-Hermitian force can only stabilize a saddle potential created by the same light that creates the non-Hermitian force, or that it could stabilize any saddle potential created otherwise.
4. For a Brownian particle trapped in a non-Hermitian force field, this system is out of equilibrium, thus the temperature should be ill-defined. Please explain how you may retain the concept of temperature in your case.
5. The number of videos is absurd! It is a chore to simply download them all. Would the authors edit them into one or two (longer) videos?

Reviewer #2:

Remarks to the Author:

Review of Non-Hermitian Non-Equipartition Theory for Trapped Particles

This article derives a theorem for energy partition of particles experiencing non-Hermitian forces. This is particularly relevant for modern physics experiments which are exploring such forces as generated by optical or acoustic traps, a subject of considerable interest. This theory is applied to several

example cases, showing some non-trivial results including unequally distributed potential energy, surprising effects of damping, and stability in a saddle potential. The paper also briefly discusses extension to multi-particle systems.

Overall, I was very impressed with the paper and recommend it be published in Nature Communications. The results are timely, and the presented theorem has many potential applications in current and future work. Although it is specifically motivated by optical and acoustic forces, non-conservative forces are currently being investigated in a growing number of areas. As a result, I believe this is a foundational result which will enable future work. It would be nice to see more development of the multi-particle theory, but I think it is reasonable to leave this to future publications. (It seems to me that this theory could also be applied to problems with multi-particle non-reciprocal forces – an area of recently growing interest in the active matter community – and the authors may want to comment on this!)

Some minor comments are listed below. Provided these issues are considered, I am strongly in favor of publication.

Minor comments:

- * A more natural title might be “Non-Hermitian Non-Equipartition Theorem for Trapped Particles,” as it is referencing what is usually referred to as the equipartition theorem.
- * Why is “potential” energy in quotes in the abstract? If it’s not exactly potential, they should elaborate.
- * The notation used for K/k could be a bit more clear. As far as I can tell, the manuscript doesn’t actually define k_{ij} as the components of $(K)_{ij}$ (although it is clear from context). However, since k usually refers to a spring constant, it would seem more natural to define equation 2 as “ $F(r) \sim -K(r - r')$ ”, where “ $(K)_{ij} = k_{ij} = -dF_i / dr_j$ ”. This would also remove the minus sign in front of the potential energy terms in eqn. 5, which seems more natural. (This would of course modify many other equations as well!)
- * “ g ” is only defined implicitly above equation 5 – it would be good to explain explicitly what g is.
- * The paper states that for the simulations the “exact” forces are computed using a Mie Scattering Theorem (page 8). It would be good to include slightly more detail here, for example, how many Mie modes were included. (Although this is a relatively standard method, it would be good to indicate the approximate accuracy of this expansion.)
- * The size of the particle used in the simulations should be indicated in the main text. It would also have been nice to see examples with several different particle sizes, as the size parameter (ka) plays a very important role in optical and acoustic forces (I believe this should affect the degree of non-Hermiticity in the force). The simulations and computations seem simple enough that this could be added with relatively little effort, and I believe it would enhance the paper.
- * On page 13: “We find that by experimentally measuring...”. Unless I missed something very important, simulations are used, not experiments. I would suggest changing to “We use simulations to numerically compute...” or something similar.
- * Top of page 12: Section title: suggest changing the “NHNE theory for multi-particle” to “multiple particles”.
- * I was unable to get any of the supplementary movies to play. On the other hand, I’m not really sure they are necessary, since statistical information about thermally excited particles is probably more relevant than time dynamics.

Reviewer #3:

Remarks to the Author:

In practice, the present MS investigates the underdamped stochastic dynamics of a Brownian particle which is subject to a thermal noise and to a non-conservative force field. This is done analytically under the assumption of a linear dependence of the force force field on the displacement of the

particle from its position of mechanical equilibrium. In particular, the MS focuses on the temporal evolution of some relevant expectation values such as those of the displacement and velocity correlation functions along the various axes, with particular emphasis on their values in the stationary state (if any). These results are then used to discuss what the authors refer to as the "non-Hermitean non-equipartition theory for trapped particles". The corresponding predictions are then illustrated in a number of cases and the accuracy of the linear approximation is tested via numerical simulations, also considering many particles in (hydrodynamic) interaction.

It seems to me that the MS contains a certain amount of unjustified jargon: for example, why do the authors introduce the notion of "non-Hermitean" force instead of referring to it non-conservative force? Why do the authors talk about non-Hermitean force matrix, while such a matrix is real and therefore it is at most non-symmetric? In which sense the main results of the MS in eq.(3) are a noteworthy generalization of the equipartition theorem? The latter takes a fairly universal form, while in the present case the final expressions of the expectation values of the quadratic observables depend significantly on the details of the potential and are therefore not universal.

After the tone of the presentation is adjusted to the actual content of the manuscript and the jargon is removed from the text, it seems to me that the remaining analysis and results, although sound, are of rather limited significance and I cannot see how they can actually move the field forward. Accordingly, I do not recommend publication in Nature Communications.

After major revisions, the manuscript might be suited for more specialized journals in, e.g., statistical physics or, in case, optics. In particular, the whole work should be put first in the proper context, the presentation revised as indicated above and the derivation of the various expressions simplified. In fact, the analysis is not particularly ingenious in many respects and it could be significantly streamlined, including the material presented in the SI. (For example, the functions in eqs.(S.13) and (S.14) are actually trivially related.)

Reviewer #1 (Remarks to the Author):

This work explores the generalized equipartition theorem of Brownian particles trapped by a non-Hermitian force field. It is a beautiful interdisciplinary study that combines non-Hermitian physics, Brownian motion, the Equipartition Theorem, and optical trapping. The authors demonstrate that in non-Hermitian force fields, the averaged energies are not evenly distributed and deviate from the expected value of $kBT/2$ per degree of freedom. The authors derive analytical formulas to describe the behavior of particles trapped by non-Hermitian force fields, which align with the Equipartition Theorem for Hermitian force fields. Additionally, the authors show that these formulas, known as the "Non-Hermitian Non-Equipartition Theory," can measure both trapping and repulsing force fields, regardless of their Hermitian or non-Hermitian nature. Numerical simulations provide nice verifications to the analytical derivations.

Reply: We thank the Reviewer #1 for the concise summary.

In my view, this manuscript addresses an important and longstanding problem from a modern perspective of non-Hermitian physics. It is likely to be of interest to both theorists and experimentalists. The presented analytical formulas not only provide insights into the behavior of particles in non-Hermitian force fields, but also offer an extension to the Equipartition Theorem. The significance of this study is further enhanced by the prevalence of non-Hermitian systems in the real world. Overall, this is an excellent piece of work with broad appeal. Nature Communications is a suitable venue. However, I do have some concerns that I would like the authors to address.

Reply: We express our gratitude to Reviewer #1 for the commendatory evaluation of our work, notably referring to it as a "beautiful interdisciplinary study". Reviewer #1's concerns are systematically addressed below.

1. It was stated multiple times in the manuscript that the theory is applicable to acoustic trapping. However, the range of particle size where acoustic trapping is applicable spanning from microscopic to macroscopic. Please be more specific on the range where the theory is applicable.

Reply: The predictions of our theoretical framework are accurate irrespective of particle size. However, in the case of macroscopic particles, such as millimeter-scale particles observed in airborne acoustic trapping, the NHNE effects are still there, albeit their magnitude may be comparatively insignificant, thus rendering observation challenging. In a non-Hermitian force field, our NHNE theory is important whenever Brownian motion is.

2. Apparently, the non-Hermitian physics is less important when ambient damping increases, for example, from low vacuum to air. Does this suggest that ambient damping counteracts the influence of non-Hermitian force fields? Is it correct to say that the theory is mainly applicable in environments with minimal ambient damping?

Reply: We have addressed the concern by including the following paragraph (red text) in the Discussion section of the main text.

“..., including in vacuum trapping applications. When there is a high-level damping, some particle behaviors predicted by our theory (such as the average energies) are very similar to what conventional ET predicts. It might seem like damping is getting rid of non-Hermiticity, but that is not the whole story. For example, the non-orthogonality of vibrational eigenmodes remains consistent, irrespective of the level of damping.”

3. Fabricating a hollow micro-sphere can be challenging. Could the author provide an alternative example that demonstrates the realization of the saddle potential? Please clarify whether the non-Hermitian force can only stabilize a saddle potential created by the same light that creates the non-Hermitian force, or that it could stabilize any saddle potential created otherwise.

Reply: The stability of a particle is governed by the linearized force matrix. Under this approximation, multiple forces of different origins can be just added together according to the superposition principle. Thus, irrespective of the sources of the saddle potential, if its strength is within the reach of optical or acoustic force, it could be stabilized.

As an additional example of saddle potential, we consider a uniform layer of dielectric coated on a gold sphere for the purpose of optical trapping, as illustrated in Figure R1.

We have included Figure R1 in Supplementary Note 9.

Fig. R1| Creating the saddle potential via coating a uniform dielectric layer on a gold sphere

in optical trapping. Components of force matrix $\vec{K}_{2D} = \begin{bmatrix} k_{xx} & g \\ -g & k_{yy} \end{bmatrix}$ (a) and its eigenvalues (b)

versus the polarization ($\hat{\mathbf{p}} = \hat{\mathbf{x}} \cos \zeta + i\hat{\mathbf{y}} \sin \zeta$) of the incident beams (with wavelength $\lambda = 1.064 \mu\text{m}$), where ζ varies from 0° to 45° . The schematic for the linear and circular polarizations are shown in (c) and (d), respectively. The particle has an inner radius $0.272 \mu\text{m}$ and an outer radius $a = 0.34 \mu\text{m}$. The refractive index of the dielectric is $n = 1.57$, while the relative permittivity of Au is $\epsilon_{\text{Au}} = -48.45 + 3.6006i$. The left-hand side of the black dashed line in (a) corresponds to the saddle potential, where $k_{xx} \cdot k_{yy} < 0$. The blue and red dashed lines in (b) indicate the neutral point

(NP), where one of the eigenvalues K_{\pm} is 0, and the exceptional point (EP), respectively. The focused beam has a numerical aperture of 0.9 and a filling factor of 1.0, and the power of each beam is normalized to 1.0 mW.

4. For a Brownian particle trapped in a non-Hermitian force field, this system is out of equilibrium, thus the temperature should be ill-defined. Please explain how you may retain the concept of temperature in your case.

Reply: Reviewer #1 is correct that the system is not in thermal equilibrium. Consequently, the temperature of the trapped particle is ill-defined. Nevertheless, the ambient temperature is still well-defined, and it is this temperature that gives Brownian motion.

By temperature in the text, we are referring to the ambient temperature. It is unchanged for transparent particles since the work done by the non-Hermitian force is on the order of $k_B T$, thus completely negligible for the entire ambient. Also, the ambient temperature is the same temperature that is being considered in the original equipartition theorem.

We have added the following red text to the main text.

“... which is fundamentally distinct from the ET. Here, we note that the temperature T refers to the ambient temperature. It is unchanged for transparent particles since the work done by the non-Hermitian force is on the order of $k_B T$ (comparable with the energy of a single molecule in the ambient), thus completely negligible for the entire ambient.”

5. The number of videos is absurd! It is a chore to simply download them all. Would the authors edit them into one or two (longer) videos?

Reply: We apologize for the inconvenience caused. We have shortened the videos and combined them all into four.

Reviewer #2 (Remarks to the Author):

Review of Non-Hermitian Non-Equipartition Theory for Trapped Particles

This article derives a theorem for energy partition of particles experiencing non-Hermitian forces. This is particularly relevant for modern physics experiments which are exploring such forces as generated by optical or acoustic traps, a subject of considerable interest. This theory is applied to several example cases, showing some non-trivial results including unequally distributed potential energy, surprising effects of damping, and stability in a saddle potential. The paper also briefly discusses extension to multi-particle systems.

Reply: We would like to express our gratitude to Reviewer #2 for providing a concise summary of our work.

Overall, I was very impressed with the paper and recommend it be published in Nature Communications. The results are timely, and the presented theorem has many potential applications in current and future work. Although it is specifically motivated by optical and acoustic forces, non-conservative forces are currently being investigated in a growing number of areas. As a result, I believe this is a foundational result which will enable future work. It would be nice to see more development of the multi-particle theory, but I think it is reasonable to leave this to future publications.

Reply: We would like to express our gratitude to Reviewer #2 for his/her very positive evaluation and for emphasizing the significance of this theoretical work.

(It seems to me that this theory could also be applied to problems with multi-particle non-reciprocal forces – an area of recently growing interest in the active matter community – and the authors may want to comment on this!)

Reply: We thank Reviewer #2 for highlighting the relevance of our work to non-reciprocal interactions and active matter, a subject of broad interest in the fields of physics and biology. Indeed, the optically coupled multiple-particle interaction resembles a type of non-reciprocal

interaction. This may enable our theory to address some non-reciprocal problems in active matter. We highlighted this by including the following remark in the main text.

“We note that the NHNE theory is of relevance to the study of non-reciprocal interaction and active matters [76-81]. These systems are typically complex, involving intricate geometries, and a variety of non-reciprocal and non-Hermitian interactions. Our theory may offer some insights and approximate predictions into these problems.”

Some minor comments are listed below. Provided these issues are considered, I am strongly in favor of publication.

Reply: We appreciate the recognition of Reviewer #2 for this work, and we have addressed the comments in a point-by-point manner.

Minor comments:

* A more natural title might be “Non-Hermitian Non-Equipartition Theorem for Trapped Particles,” as it is referencing what is usually referred to as the equipartition theorem.

Reply: We understand that the suggestion of Reviewer #2 is very reasonable. Yet, after we discussed among ourselves, we still preferred the “NHNE theory” which is more encompassing. We added the following note (red text) in the main text to highlight the equivalence of the two terms:

“...thereby generalize the ET to the non-Hermitian non-equipartition (NHNE) theory (may be equivalently termed non-Hermitian non-equipartition theorem).”

* Why is “potential” energy in quotes in the abstract? If it’s not exactly potential, they should elaborate.

Reply: We thank Reviewer #2 for drawing this matter to our attention. Since the total force is nonconservative, it cannot be derived from the gradient of potential energy. In this sense, no potential energy can be defined for the entire system, and this is why we quote the term potential.

Nevertheless, the conservative part of the force:

$$\mathbf{F}_{\text{conservative}} = \left(\frac{\vec{\mathbf{K}} + \vec{\mathbf{K}}^T}{2} \right) \cdot \Delta \mathbf{x} = -\nabla U ,$$

is derivable from a potential energy

$$U = -\left[\Delta \mathbf{x} \cdot \left(\vec{\mathbf{K}} + \vec{\mathbf{K}}^T \right) \cdot \Delta \mathbf{x} \right] / 4 .$$

So the term potential energy still has its physical meaning.

We have removed the quotes from the Abstract, and we have delineated the issue in the main text by rewriting the following sentence (red text):

“... Crucially, the last two quantities, referred to as potential energies, only represent the conservative interaction, but not the entire non-Hermitian interactions. The latter cannot be expressed as the gradient of a potential.”

* The notation used for K/k could be a bit more clear. As far as I can tell, the manuscript doesn't actually define k_{ij} as the components of $(K)_{ij}$ (although it is clear from context).

Reply: We agree with the Reviewer #2. We have now explicitly defined the force matrix components as $\vec{\mathbf{K}}_{ij} = k_{ij} = \frac{\partial F_i}{\partial r_j}$ in the main text.

However, since k usually refers to a spring constant, it would seem more natural to define equation 2 as “ $F_i \sim -K_{ij} (r_i - r_j)$ ”, where “ $(K)_{ij} = k_{ij} = -dF_i / dr_j$ ”. This would also remove the minus sign in front of the potential energy terms in eqn. 5, which seems more natural. (This would of course modify many other equations as well!)

Reply: We agree with Reviewer #2. The notation $\mathbf{F}(\mathbf{r}) \approx \vec{\mathbf{K}} \cdot (\mathbf{r} - \mathbf{r}')$ in Eq. (2) has been consistently used by some of the authors of the current manuscript for nearly 20 years [e.g., Phys. Rev. B 72, 085130 (2005); PRL 104, 103601 (2010); Nat. Commun. 12 (1), 6597 (2021)]. The convention suggested by Reviewer #2 is more natural indeed, but consistency is also very

important. After careful discussion, we decided to retain the original notation for consistency with previous works. We appreciate the feedback from Reviewer #2 in bringing this to our attention.

* “g” is only defined implicitly above equation 5 – it would be good to explain explicitly what g is.

Reply: We have included the following explanation (red text) of g in the main text.

“... is a rotation matrix. Here, the existence of the anti-symmetric component g in the force matrix is a consequence of non-Hermiticity. One can infer from the force field produced by g that this nonconservative force revolves the particle around the mechanical equilibrium.”

* The paper states that for the simulations the “exact” forces are computed using a Mie Scattering Theorem (page 8). It would be good to include slightly more detail here, for example, how many Mie modes were included. (Although this is a relatively standard method, it would be good to indicate the approximate accuracy of this expansion.)

Reply: We thank Reviewer #2 for the careful review. We include the following sentence (red text) in the main text.

“... instead of the linearized force based on the force matrix. In the Mie scattering theory, the expansion series are truncated at $L_{\max} = ka + 4(ka)^{1/3} + 2$ [69,70], with $k = \frac{2\pi}{\lambda}$ being the wavenumber. In some cases, we verified the convergence of our calculations by comparing the calculation truncated at L_{\max} with those truncated at $L_{\max} + 5$.”

* The size of the particle used in the simulations should be indicated in the main text.

Reply: We have now provided all the particle sizes considered in the main text.

It would also have been nice to see examples with several different particle sizes, as the size parameter (ka) plays a very important role in optical and acoustic forces (I believe this should affect the degree of non-Hermiticity in the force). The simulations and computations seem simple enough that this could be added with relatively little effort, and I believe it would enhance the paper.

Reply: We agree with Reviewer #2. The particle radius (a) indeed affects the non-Hermiticity significantly. We conducted new simulations for a dielectric particle ($n = 1.57$), as depicted in Figure R2a. The components and eigenvalues of the force matrix are presented in Figure R2b and R2c, respectively, while varying the particle radius from $a = 0.1$ to $0.6 \mu\text{m}$.

The trapping stiffnesses k_{xx} and k_{yy} increase roughly monotonically, however, the non-Hermitian and nonconservative component g does not. According to Figure R2b, g is small for a small radius, owing to the weak scattering force associated with Rayleigh particles. g is relatively large for $a = 0.3$ to $0.5 \mu\text{m}$, and then drops again.

Figures R2d-e and R2f-g display the kinetic and potential energies, respectively, for different viscosities: $\eta = 3.0 \mu\text{Pa}\cdot\text{s}$ and $\eta = 18.4 \mu\text{Pa}\cdot\text{s}$ (Air). It is worth noting that in addition to optical force, the particle mass, which is proportional to the particle radius, also has a significant impact (as depicted by Eq. (5) in the main text).

We have included Figure R2 in Supplementary Note 8.

Fig. R2| Influence of particle radius on the non-Hermitian non-equipartitioning in optical trapping. (a) A schematic showing a dielectric Brownian particle with a refractive index $n = 1.57$

is optically trapped by two counter-propagating focused beams (with wavelength $\lambda = 1.064 \mu\text{m}$ and circular polarization). The focused beam has a numerical aperture of 0.9 and a filling factor of 1.0, and the beam power for each beam is normalized to 1.0 mW. (b) Components of the force

matrix $\vec{\mathbf{K}}_{2\text{D}} = \begin{bmatrix} k_{xx} & g \\ -g & k_{yy} \end{bmatrix}$ and (c) its eigenvalues for different radius a , ranging from $0.1 \mu\text{m}$ to 0.6

μm . For different viscosities, $\eta = 3.0 \mu\text{Pa}\cdot\text{s}$ (d–e) and $\eta = 18.4 \mu\text{Pa}\cdot\text{s}$ (f–g), average energies

$\frac{1}{2}m\langle v_x^2 \rangle$, $\frac{1}{2}m\langle v_y^2 \rangle$, $-\frac{1}{2}k_{xx}\langle x^2 \rangle$, and $-\frac{1}{2}k_{yy}\langle y^2 \rangle$ are presented as lines.

* On page 13: “We find that by experimentally measuring...”. Unless I missed something very important, simulations are used, not experiments. I would suggest changing to “We use simulations to numerically compute...” or something similar.

Reply: We apologize for the misleading statements. We corrected our statement as: “We **propose** that by experimentally measuring...”.

* Top of page 12: Section title: suggest changing the “NHNE theory for multi-particle” to “multiple particles”.

Reply: We have implemented the suggestion.

* I was unable to get any of the supplementary movies to play. On the other hand, I’m not really sure they are necessary, since statistical information about thermally excited particles is probably more relevant than time dynamics.

Reply: We apologize for the inconvenience caused, and we have shortened the video and combined them into four.

Reviewer #3 (Remarks to the Author):

We thank Reviewer #3 for reviewing our manuscript. The concerns raised by Reviewer #3 are addressed in a point-by-point manner below.

In practice, the present MS investigates the underdamped stochastic dynamics of a Brownian particle which is subject to a thermal noise and to a non-conservative force field.

Reply: One of the main themes of our work is also the application of non-Hermitian physics to optical trapping, which is a non-Hermitian system. Non-Hermitian systems and optical trapping in vacuum or air (which are underdamped) hold significant and growing importance.

The manuscript has been modified to emphasize this aspect (red text).

“...the application of non-Hermitian physics to Brownian dynamics. **Specifically, one of our main themes is the application of non-Hermitian physics to optical trapping and binding, which are non-Hermitian systems. Non-Hermitian systems and optical trapping in vacuum or air (which are underdamped) hold significant and growing importance [29-31,58,59,61,62].**”

This is done analytically under the assumption of a linear dependence of the force force field on the displacement of the particle from its position of mechanical equilibrium. In particular, the MS focuses on the temporal evolution of some relevant expectation values such as those of the displacement and velocity correlation functions along the various axes, with particular emphasis on their values in the stationary state (if any). These results are then used to discuss what the authors refer to as the "non-Hermitean non-equipartition theory for trapped particles". The corresponding predictions are then illustrated in a number of cases and the accuracy of the linear approximation is tested via numerical simulations, also considering many particles in (hydrodynamic) interaction.

Reply: We would like to express our gratitude to Reviewer #3 for summarizing our research. The "assumption of a linear dependence of the force field" is relevant and applicable to trapping, which is a phenomenon widely observed in various domains, including optical and acoustic trapping,

among others. Hence, we believe that our framework holds significant relevance across a wide range of applications.

It seems to me that the MS contains a certain amount of unjustified jargon: for example, why do the authors introduce the notion of "non-Hermitean" force instead of referring to it non-conservative force?

Reply: In the context of our work, it is crucial to describe the force as 'non-Hermitian' to align with the wider field of non-Hermitian physics. Here, the non-conservative force we are considering can be represented by a non-Hermitian force matrix, but not all non-conservative forces can be. Within the realm of non-Hermitian physics, non-Hermitian matrices are renowned for their significant properties, such as the exceptional point that indicates a transition from stability to instability. Furthermore, we have previously utilized non-Hermitian theory to demonstrate the inherent instability of a many-particle system [Nat. Commun. 12 (1), 6597 (2021)].

By using the term “non-conservative”, which encompasses non-Hermitian systems, we run the risk of obscuring the link to non-Hermitian physics and, as a result, the clarity of the underlying physics.

Why do the authors talk about non-Hermitean force matrix, while such a matrix is real and therefore it is at most non-symmetric?

Reply: Our force matrix can be described either as real non-symmetric or non-Hermitian. It is crucial to refer to it as 'non-Hermitian' in this context because the principles of non-Hermitian physics apply to all non-Hermitian matrices, including real non-symmetric ones. By using the term 'non-Hermitian,' we not only emphasize its connection to and significance within non-Hermitian physics but also highlight the fascinating mathematics and physics inherent in the system.

In which sense the main results of the MS in eq.(3) are a noteworthy generalization of the equipartition theorem?

Reply: The Equipartition Theorem is inadequate when applied to non-Hermitian trapping forces. Our theory addresses this limitation by extending the Equipartition Theorem to encompass both Hermitian and non-Hermitian force fields. Importantly, in situations involving Hermitian forces, our theory aligns with the conventional Equipartition Theorem. Considering the importance of non-Hermitian systems (see also the comments from Reviewer #1 and Reviewer #2), we posit that our theory offers a valuable expansion of the Equipartition Theorem.

We have included the following sentence (red text) in the main text to emphasize the generalization of the NHNE theory.

“...that it represents is referred to as the NHNE theory. We note that the NHNE theory, denoted by Eq. (3), can be applied to any force matrix $\vec{\mathbf{K}}$, including both Hermitian and non-Hermitian matrices, for a trapped Brownian particle.”

The latter takes a fairly universal form,

Reply: Our theory simplifies to the Equipartition Theorem when applied to Hermitian trapping forces. Consequently, if the Equipartition Theorem is 'fairly universal,' our theory can be seen as having an even wider range of applications.

while in the present case the final expressions of the expectation values of the quadratic observables depend significantly on the details of the potential and are therefore not universal.

Reply: To apply the originally equipartition theorem, every single element (2 independent elements in total) of the 2D force matrix is needed. Similarly, to apply for our NHNE theory, every single element (3 independent elements in total) is needed.

We do not agree with the idea that our theory "depends significantly on the details of the potential and is therefore not universal." Similar to the Equipartition Theorem, our theory takes into account all elements in the force matrix. Hence, the universality of our theory should not be doubted.

After the tone of the presentation is adjusted to the actual content of the manuscript and the jargon is removed from the text,

Reply: As stated in the above replies, explanations are now incorporated into the main text to clarify the necessity of using these terms.

it seems to me that the remaining analysis and results, although sound,

Reply: We appreciate the reviewer's recognition of the soundness of our work.

are of rather limited significance and I cannot see how they can actually can move the field forward. Accordingly, I do not recommend publication in Nature Communications.

After major revisions, the manuscript might be suited for more specialized journals in, e.g., statistical physics or, in case, optics.

Reply: The Equipartition Theorem holds significant interdisciplinary importance, transcending traditional boundaries between scientific disciplines. Its applications span across physics, chemistry, biology, engineering, and even fields like materials science. The interdisciplinary nature of the Equipartition Theorem underscores its wide-ranging impact on diverse scientific domains. In this paper, we have generalized the Equipartition Theorem to NHNE theory, which considers non-Hermitian force fields. These force fields can be optical forces, acoustic forces, and mechanical forces. Therefore, we anticipate that the NHNE theory will have broader applications in fields where the Equipartition Theorem has played significant roles. Furthermore, we would like to express our gratitude to Reviewer #2 for highlighting the close relationship between this study and the field of active matter, which has garnered significant attention from the disciplines of physics and biology. As a result, we believe it would be inappropriate to categorize this work as “of limit significance” and “specialized in statistical physics or optics”.

In particular, the whole work should be put first in the proper context, the presentation revised as indicated above and the derivation of the various expressions simplified. In fact, the analysis is not

particularly ingenious in many respects and it could be significantly streamlined, including the material presented in the SI. (For example, the functions in eqs.(S.13) and (S.14) are actually trivially related.)

Reply: We have reviewed the derivations and formulas presented in both the main text and Supplementary Notes, and they have already been simplified.

Eq.(S13)-(S14) mentioned by Reviewer #3, are in fact definitions that serve to simplify the latter formula. In our opinion, a definition cannot be classified as “trivially (or non-trivially) related.”

Having addressed all the concerns raised, we hope the reviewer will reconsider his/her initial perspective.

Reviewers' Comments:

Reviewer #1:

Remarks to the Author:

The authors have properly addressed my comments.

I think this is an excellent and novel work and I highly recommend its publication.

Reviewer #2:

Remarks to the Author:

As described in my original review, I believe this is a novel and well written manuscript which is worthy of publication in Nature Communications. I had offered some minor suggestions, and the majority of these have been addressed. Although in some cases they have left the manuscript unchanged, I am satisfied for their rationale for doing so. In particular I would note that they have clarified all of the ambiguities I identified in the original manuscript (i.e. definitions of various things, the precise details of the Mie calculations, etc.). I was also very pleased to see the supplemental figure on the influence of particle radius, as this is very relevant for experimentalists who would wish to realize these effects in real systems. The new version of the supplemental movies also play just fine for me now -- I'm not sure what the previous issue was but it has been corrected.

In short: these revisions have improved an already worthy manuscript. I recommend publishing the current version as is.

Reviewer #3:

Remarks to the Author:

The Authors have considered the points I listed in my previous report. However, I cannot actually say that I am satisfied by their replies. In fact, I still think that the notion of "non-Hermitean physics" is appropriate in a quantum context where one might want to consider an Hamiltonian which is not an Hermitean operator. In the context of classical physics, relevant for the present MS, instead, this seems to be an unnecessary jargon. The very fact that the authors used the notion of "non-Hermitean forces" in previous works is not really a convincing argument to keep on using it. Concerning the notion of universality, I think that the authors misunderstood what I meant. In fact, in statistical physics, a statement is considered to have a certain degree of universality if it is largely independent of the actual values of the parameters which define the system. In this sense, the equipartition theorem is a universal statement because the fact that the expectation value of any quadratic contribution (involving the canonical variables) in the Hamiltonian is $k_B T/2$ is true irrespective of the details of the system. In its "non-Hermitean" version, this is no longer true: in order to know the dimensionless factor which multiplies $k_B T/2$ while taking the averages of quadratic observables one needs to know the details of the system. Universality is a concept which has nothing to do with the idea of "wide applicability of the result" which the authors are seemingly referring to in their reply.

This said, my opinion on the manuscript does not change after revision and after having read the reports of the other two Referees.

In fact, I still fail to appreciate the sense in which this manuscript provides a fundamental contribution to our understanding of physical system, but this might well be due to the fact that I probably belong to a different community than the authors and the other two Referees. I am also still convinced that the derivation of the main result of the paper can be significantly simplified. After all, the authors are solving a second-order linear but non-homogeneous differential equation which, in the possible stationary state even simplifies further. Concerning (S.13) and (S.14), please note that one function is

the derivative of the other and therefore I still do not see why different symbols should be used, especially due to the fact that they do not have a direct physical meaning which makes it worth distinguish them.

Reviewer #1 (Remarks to the Author):

The authors have properly addressed my comments.

I think this is an excellent and novel work and I highly recommend its publication.

Reply: We express our gratitude to Reviewer #1 for recommending the publication of this research work.

Reviewer #2 (Remarks to the Author):

As described in my original review, I believe this is a novel and well written manuscript which is worthy of publication in Nature Communications. I had offered some minor suggestions, and the majority of these have been addressed. Although in some cases they have left the manuscript unchanged, I am satisfied for their rationale for doing so. In particular I would note that they have clarified all of the ambiguities I identified in the original manuscript (i.e. definitions of various things, the precise details of the Mie calculations, etc.). I was also very pleased to see the supplemental figure on the influence of particle radius, as this is very relevant for experimentalists who would wish to realize these effects in real systems. The new version of the supplemental movies also play just fine for me now -- I'm not sure what the previous issue was but it has been corrected.

In short: these revisions have improved an already worthy manuscript. I recommend publishing the current version as is.

Reply: We express our gratitude to Reviewer #2 for his/her favorable feedback and authorization to publish our research.

Reviewer #3 (Remarks to the Author):

We thank Reviewer #3 for his/her review and clarifications.

We would like to clarify that our intention was never to imply that our discovery holds the same level of importance as the original equipartition theorem. We deeply respect the universality of the original equipartition theorem and its groundbreaking impact on the field of physics during its time. This is one reason why we call our work “Non-Hermitian Non-Equipartition Theory” instead of “Non-Hermitian Non-Equipartition Theorem.”

On the other hand, our work represents a significant advancement by extending the equipartition theorem to the non-Hermitian regime (the use of the term “non-Hermitian” in classical physics is justified later). The demand for a non-Hermitian theory applicable to trapped Brownian particles is indeed substantial and compelling. The potential applications of such a theory encompass various domains, including but not limited to optical trapping, acoustic trapping, and active matter. This can be partially seen from the responses of the other two reviewers.

We explicitly addressed modern non-Hermitian (and non-reciprocal interaction) problems that hold significant importance in contemporary contexts. We provided a new quantitative method to effectively address these challenges. Our theory adequately meets the urgent need for a viable solution and contributes significantly to the understanding and resolution of these issues. More details about the importance of our work are provided in the later parts.

The Authors have considered the points I listed in my previous report. However, I cannot actually say that I am satisfied by their replies.

Reply: We hope that our new reply can adequately address Reviewer #3’s concerns.

In fact, I still think that the notion of "non-Hermitean physics" is appropriate in a quantum context where one might want to consider an Hamiltonian which is not an Hermitean operator. In the context of classical physics, relevant for the present MS,

instead, this seems to be an unnecessary jargon. The very fact that the authors used the notion of "non-Hermitian forces" in previous works is not really a convincing argument to keep on using it.

Reply: According to Reviewer #3, the use of the term “non-Hermitian” is inappropriate for the current manuscript, and “non-Hermitian” should be reserved for quantum mechanical phenomena. Indeed, non-Hermitian physics has a history rooted in quantum mechanics, and it has great applications there.

It is worth noting that over the past two decades and beyond, a substantial body of work in classical physics (including optics, acoustic, mechanics, circuits, electromagnetics) has emerged under the label of non-Hermitian physics. The field is collectively known as non-Hermitian physics, which is currently quite “hot.” We have compiled over 20 *review* papers (by no means exhaustive) related to classical non-Hermitian physics to support this claim, some can be dated back to 25 years ago. Many of these papers have been published in highly regarded journals such as Nature Reviews Physics, Nature Nanotechnology, Nature Physics, Reviews of Modern Physics, and Science.

Some review papers about non-Hermitian classical physics:

-
1. Ching, E. S. C. et al. Quasinormal-mode expansion for waves in open systems. *Rev. Mod. Phys.* 70, 1545–1554 (1998).
 2. El-Ganainy, R. et al. Non-Hermitian physics and PT symmetry. *Nature Phys* 14, 11–19 (2018).
 3. Miri, M.-A. & Alù, A. Exceptional points in optics and photonics. *Science* 363, eaar7709 (2019).
 4. Gupta, S. K. et al. Parity-Time Symmetry in Non-Hermitian Complex Optical Media. *Advanced Materials* 32, 1903639 (2020).
 5. Ashida, Y., Gong, Z. & Ueda, M. Non-Hermitian physics. *Advances in Physics* 69, 249–435 (2020).
 6. Wiersig, J. Review of exceptional point-based sensors. *Photon. Res., PRJ* 8, 1457–1467 (2020).

7. Bergholtz, E. J., Budich, J. C. & Kunst, F. K. Exceptional topology of non-Hermitian systems. *Rev. Mod. Phys.* 93, 015005 (2021).
8. Wang, H. et al. Topological physics of non-Hermitian optics and photonics: a review. *J. Opt.* 23, 123001 (2021).
9. Parto, M., Liu, Y. G. N., Bahari, B., Khajavikhan, M. & Christodoulides, D. N. Non-Hermitian and topological photonics: optics at an exceptional point. *Nanophotonics* 10, 403–423 (2021).
10. Gu, Z. et al. Controlling Sound in Non-Hermitian Acoustic Systems. *Phys. Rev. Appl.* 16, 057001 (2021).
11. Zhang, X., Zhang, T., Lu, M.-H. & Chen, Y.-F. A review on non-Hermitian skin effect. *Advances in Physics: X* 7, 2109431 (2022).
12. Ding, K., Fang, C. & Ma, G. Non-Hermitian topology and exceptional-point geometries. *Nat Rev Phys* 4, 745–760 (2022).
13. De Carlo, M., De Leonardis, F., Soref, R. A., Colatorti, L. & Passaro, V. M. N. Non-Hermitian Sensing in Photonics and Electronics: A Review. *Sensors* 22, 3977 (2022).
14. Hurst, H. M. & Flebus B. Non-Hermitian physics in magnetic systems. *J. Appl. Phys.* 132, 220902 (2022).
15. Okuma, N. & Sato, M. Non-Hermitian Topological Phenomena: A Review. *Annu. Rev. Condens. Matter Phys.* 14, 83–107 (2023).
16. Wang, C. et al. Non-Hermitian optics and photonics: from classical to quantum. *Adv. Opt. Photon., AOP* 15, 442–523 (2023).
17. Lin, R., Tai, T., Li, L. & Lee, C. H. Topological non-Hermitian skin effect. *Front. Phys.* 18, 53605 (2023).
18. Li, A. et al. Exceptional points and non-Hermitian photonics at the nanoscale. *Nat. Nanotechnol.* 18, 706–720 (2023).
19. Banerjee, A., Sarkar, R., Dey, S. & Narayan, A. Non-Hermitian topological phases: principles and prospects. *J. Phys.: Condens. Matter* 35, 333001 (2023).
20. Yan, Q. et al. Advances and applications on non-Hermitian topological photonics. *Nanophotonics* 12, 2247–2271 (2023).

21. Wang, X., Dong, R., Li, Y. & Jing, Y. Non-local and non-Hermitian acoustic metasurfaces. *Rep. Prog. Phys.* 86, 116501 (2023).
 22. Huang, L. et al. Acoustic resonances in non-Hermitian open systems. *Nat Rev Phys* 1–17 (2023).
-

In classical physics, the term "non-Hermitian physics" has been extensively used, and our work expands the realm of non-Hermitian physics to include Brownian motion and trapping phenomena. We believe that the term "non-Hermitian," originally a pure mathematical concept, can be applied to both classical and quantum contexts, and it has been broadly applied in both areas already. Thus, the term “non-Hermitian” is also appropriate for classical physics and our manuscript. In fact, since our initial work on non-Hermitian optical trapping and binding [*Nature Commun.* **12**(1), 6597 (2021)], the term “non-Hermitian” is spreading in the field of optical micromanipulation. We have compiled a non-exhaustive list of “non-Hermitian” papers in optical trapping and binding to support this claim. We believe that the term “non-Hermitian” will eventually be planted into optical trapping and binding, and become an important part of it.

Some research works follow our non-Hermitian optical micromanipulation theory. The non-Hermitian theory is spreading.

23. Yu, X., Jin, Y., Shen, H., Han, Z. & Zhang, J. Hermitian and non-Hermitian normal-mode splitting in an optically-levitated nanoparticle. *Quantum Front* 1, 6 (2022).
24. Brzobohatý, O. et al. Synchronization of spin-driven limit cycle oscillators optically levitated in vacuum. *Nat Commun* 14, 5441 (2023).
25. Yokomizo, K. & Ashida, Y. Non-Hermitian physics of levitated nanoparticle array. *Phys. Rev. Research* 5, 033217 (2023).
26. Rudolph, H., Delić, U., Hornberger, K. & Stickler, B. A. Quantum theory of non-Hermitian optical binding between nanoparticles. Preprint at <https://doi.org/10.48550/arXiv.2306.11893> (2023).
27. Reisenbauer, M. et al. Non-Hermitian dynamics and nonreciprocity of optically coupled nanoparticles. Preprint at <https://doi.org/10.48550/arXiv.2310.02610> (2023).

28. Liška, V. et al. Observations of a PT-like phase transition and limit cycle oscillations in non-reciprocally coupled optomechanical oscillators levitated in vacuum. Preprint at <https://doi.org/10.48550/arXiv.2310.03701> (2023).

In the manuscript, we have already emphasized the importance of non-Hermitian physics in both quantum and classical physics. Here, we have made modifications and additions (highlighted in green) to further highlight its significance specifically in classical physics:

Page 3

“In recent years, interest in non-Hermitian physics [33,34] has grown. This interest was initially sparked by studies in quantum mechanics [35-39] and then spread to a wide range of areas of physical science, including classical mechanics [40,41], optics [42-44], acoustics [45,46], metamaterials [47,48], electrical circuits [49-52], nuclear magnetic resonance [53], topological photonics [54-56], and optical manipulation [31,57-62]. We note that many of these topics are associated with classical physics. Here, we explore the application of non-Hermitian physics in Brownian dynamics. Specifically, one of our main themes is the application of non-Hermitian physics to optical trapping and binding, which are non-Hermitian systems. Non-Hermitian systems and optical trapping in vacuum or air (which are underdamped) hold significant and growing importance [29-31,58-62].”

Concerning the notion of universality, I think that the authors misunderstood what I meant. In fact, in statistical physics, a statement is considered to have a certain degree of universality if it is largely independent of the actual values of the parameters which define the system. In this sense, the equipartition theorem is a universal statement because the fact that the expectation value of any quadratic contribution (involving the canonical variables) in the Hamiltonian is $k_B T/2$ is true irrespective of the details of the system. In its "non-Hermitian" version, this is no longer true: in order to know the dimensionless factor which multiplies $k_B T/2$ while taking the averages of quadratic observables one needs to know the details of the system. Universality is a concept which

has nothing to do with the idea of "wide applicability of the result" which the authors are seemingly referring to in their reply.

Reply: We express our gratitude to Reviewer #3 for pointing out the universality of the original equipartition theorem. We agree with Reviewer #3 regarding the universality and significance of the original equipartition theorem. However, we would like to emphasize that this does not diminish the significance of our study. Specifically, our theory offers an analytical theory for addressing Brownian dynamics within a non-Hermitian force field. With our study (NHNE theory), we can predict the partitioned kinetic or potential energies for each degree of freedom of micro-particles trapped by non-Hermitian force fields. In the special case of no non-Hermitian forces, our theory reduces to the original equipartition theorem, thus it has a wider applicability. This is clearly of importance to the optical trapping and non-Hermitian communities.

In the **Discussion**, we have made the following modifications and additions (highlighted in green) to clarify the relationship between our NHNE theory and the original equipartition theorem accordingly:

Page 14

“In this article, stochastic calculus is applied to generalize the ET to deal with non-Hermitian trapping and binding forces. This generalized theory, denoted by the NHNE theory, enables the calculation of the average energies of a single particle or a group of trapped or bounded particles, even when the force matrix of a system is non-Hermitian. By "generalize," we meant to extend the original ET to address non-Hermitian problems. We note that the NHNE theory reveals the breaking of universality in the original ET by non-Hermiticity, in the sense that the average energies associated with each degree of freedom are no longer equal and depends on the details of the system. This is a new development in the study of Brownian motion and has far-reaching implications for a variety of problems associated with modern technology, including those associated with optical/acoustic trapping and binding, and with other open mechanical systems.”

This said, my opinion on the manuscript does not change after revision and after having read the reports of the other two Referees.

In fact, I still fail to appreciate the sense in which this manuscript provides a fundamental contribution to our understanding of physical system, but this might well be due to the fact that I probably belong to a different community than the authors and the other two Referees.

Reply: We hope Reviewer #3 can reconsider the wide applicability of our theory and recognize the importance of the non-Hermitian mechanical system that we have presented in our manuscript. In addition to what is discussed above, we demonstrate how our NHNE theory can be used to measure both Hermitian and non-Hermitian force fields by statistically analyzing the trajectories of the manipulated particles subjected to optical force, acoustical force, mechanical force, or other forces. We propose a novel method for measuring the repulsing force exerted on a microparticle using a non-Hermitian force field. This is completely different from using a "spring force gauge" to measure the force on a macroscopic scale. Due to the nature of the repulsing force field, the microparticle tends to escape, making it challenging to measure this force at a microscopic scale. However, by employing the non-Hermitian force field to stabilize the microparticle, we can extract the repulsing force field information from the statistical data of the trapped particle trajectories, aided by the NHNE theory.

These potential applications highlight the significance of our study, and we sincerely hope Reviewer #3 will reconsider the importance of our findings, which contribute fruitfully to the relevant physical communities.

In the **Discussion**, we have made the following modifications and additions (highlighted in green) to emphasize the applicability of our theory and its fundamental contribution to physical systems:

Page 15

“To provide a concrete illustration of the NHNE theory, we focus on optical trapping, which is one example of a non-Hermitian trapping system. The NHNE theory is applied to analyze both the trapping potential (Fig. 1a) and the saddle potential (Fig. 3a). We

propose that by experimentally measuring $\langle x^2 \rangle$, $\langle y^2 \rangle$, $\langle xy \rangle$, $\langle v_x^2 \rangle$, $\langle v_y^2 \rangle$, and $\langle v_x v_y \rangle$, Eq. (5) can be used to determine both the Hermitian (k_{xx} and k_{yy}) and non-Hermitian (g) force constants. To our knowledge, such an approach is previously limited by the availability of pertinent theories, as a result, they can only be used to perform such indirect measurement of force constants in a heavily damped environment [57,75]. Thus, the NHNE theory provides a novel method to directly measure force constants under arbitrary levels of damping, including in vacuum trapping applications. When there is a large damping, some predictions by our theory (such as the average energies) can be very similar to what conventional ET predicts. It might seem like damping is getting rid of non-Hermiticity, but that is not the complete physical picture. For example, the vibrational eigenmodes remains non-orthogonality, irrespective of the damping. Moreover, we make the surprising finding that non-Hermitian forces can stabilize a particle in a saddle potential. Repulsive forces at the microscopic scale can be difficult to measure, due to the absence of a stable equilibrium. We propose that a microparticle located near a saddle potential can be stabilized by non-Hermitian forces. Furthermore, the repulsive force constant in a saddle potential can be determined using the NHNE theory.”

I am also still convinced that the derivation of the main result of the paper can be significantly simplified.

Reply: We appreciate Reviewer #3’s suggestions, which are aimed to make the Supplementary Information more concise. In our opinion, it is important to consider from the perspective of potential readers, and in this case, we expect a broad readership. Some of them (if not most) may not be familiar with stochastic calculus. The readers typically turn to the Supplementary Information for additional details and guidance on deriving the formulas. This allows them to grasp and potentially build upon this research in relation to other pertinent subjects. The comprehensive derivations in the Supplementary Information will undeniably assist readers with diverse or tangential backgrounds in swiftly comprehending the underlying physics. Consequently, this

would allow the study to garner broader interest.

In our opinion, the derivations in the Supplementary Information are suitable for readers with different backgrounds. In addition, our derivations follow the methodology outlined in the renowned review paper by Chandrasekhar, S. titled "Stochastic problems in physics and astronomy" [Rev. Mod. Phys. **15**(1), 1, 1943]. In this paper, the author utilized 34 equations (Eq. (184-217) in the review paper) to derive the result for the case of a 1-dimensional Hermitian force field. In our work, we used 35 equations (Eq. (8-42) in our Supplementary Information) to illustrate our derivation. It is worth noting that we are dealing with a more complex 2-dimensional non-Hermitian force field. Furthermore, our derivations include the calculation of ensemble averages, which were not present in the original 34 equations in the Review of Modern Physics paper. This suggests that the length of derivation is not significantly longer than required.

To summarize, we believe the inclusion of derivation steps as presented in the Supplementary Information is crucial. These steps provide essential insights and clarity to support our findings.

After all, the authors are solving a second-order linear but non-homogeneous differential equation which, in the possible stationary state even simplifies further.

Reply: To be more precise, we are solving a *stochastic differential equation* (SDE), which is typically difficult and tedious. In our case, the SDE involves both deterministic and stochastic forces, which makes them challenging. The particle trajectories under the stochastic forces are not repeatable even under the same initial conditions. This is why we typically calculate the ensemble averages of specific physical quantities.

It turns out that the SDE in our non-Hermitian physical model can be solved analytically. After solving the SDE, we calculate the ensemble averages considering all possible trajectories. This approach allows us to derive a formalism that not only provides steady-state solutions but also captures the non-steady state transient solutions. These transient solutions are significant components of our manuscript and have their significance in their own right.

Concerning (S.13) and (S.14), please note that one function is the derivative of the other and therefore I still do not see why different symbols should be used, especially due to the fact that they do not have a direct physical meaning which makes it worth distinguish them.

Reply: We thank Reviewer #3 for his/her clarification. In fact, Eqs. (S.13) and (S.14) adhere to the notation used in reference [Chandrasekhar, S. Stochastic problems in physics and astronomy. *Rev. Mod. Phys.* **15**(1), 1 (1943).], specifically in its Eq. (199). Using similar notation can help establish a link between our paper and the literature.

We agree that some of these definitions may not have direct physical meanings, but these notations have provided us with simplified equations. For instance, Eqs. (S.16), (S.18), (S.19), (S.23), (S.24), (S.29), (S.30), (S.33), (S.34), (S.35), (S.37), and (S.40) have been significantly simplified using Eqs. (S.13) and (S.14). Moreover, the notations in Eqs. (S.13) and (S.14) have also facilitated more concise derivations for the higher-dimensional NHNE theory, as demonstrated in Eqs. (S.43)-(S.47).

Reviewers' Comments:

Reviewer #3:

Remarks to the Author:

The authors have addressed carefully all the points I raised in my previous reports and they have revised the text in order to reduce possible misunderstanding and clarify the context of their findings. In this sense, the overall text has been improved. They have also provided extensive evidence of the fact that some of the terms that I was considering as "jargon" are actually widely used in a community which clearly I do not belong to.

Concerning my proposal to simplify the derivation of their final results, the authors argued that in an influential paper of the late '40s they did the same (so what?) and that they prefer to keep the derivation as it is for pedagogical reasons.

Overall, I certainly appreciate the effort the authors did in order to try to change my opinion on their MS.

However, I cannot say that they they have been successful.

On the other hand, the enthusiasm of the other two Referees makes me wondering if I am not misjudging the impact of this work.

Reviewer #3 (Remarks to the Author):

The authors have addressed carefully all the points I raised in my previous reports and they have revised the text in order to reduce possible misunderstanding and clarify the context of their findings. In this sense, the overall text has been improved. They have also provided extensive evidence of the fact that some of the terms that I was considering as "jargon" are actually widely used in a community which clearly I do not belong to.

Reply: We express our gratitude to Reviewer #3 for his/her review.

Concerning my proposal to simplify the derivation of their final results, the authors argued that in an influential paper of the late '40s they did the same (so what?) and that they prefer to keep the derivation as it is for pedagogical reasons.

Reply: We thank the reviewer for his suggestion. As already explained in previous communications, the notation adopted by us is similar to that RMP paper [Rev. Mod. Phys. 15(1), 1, 1943]. We believe that following this convention can improve the readability of our derivation. Nevertheless, we stress that despite the notations are similar, the problems we solved are not the same. We solve the harder non-Hermitian multi-dimensional problem, in contrast to the Hermitian one-dimension problem solved by the RMP paper.

“We stress that the problem we resolved is a multi-dimensional non-Hermitian problem within the framework of the Langevin equation. In contrast, owing to the Hermiticity, the 3 cartesian directions are independent in Ref. [i], therefore the problem reduced to a single dimension one.”

Overall, I certainly appreciate the effort the authors did in order to try to change my opinion on their MS.

However, I cannot say that they they have been successful.

On the other hand, the enthusiasm of the other two Referees makes me wondering if I am not misjudging the impact of this work.

Reply: After addressing the concerns of Reviewer #3, we hope that the revised version will demonstrate the significance of our research more effectively.